# Cell-Fate Determination from Embryo to Cancer Development: Genomic Mechanism Elucidated

**DOI:** 10.3390/ijms21134581

**Published:** 2020-06-27

**Authors:** Masa Tsuchiya, Alessandro Giuliani, Kenichi Yoshikawa

**Affiliations:** 1SEIKO Life Science Laboratory, SRI, Osaka 540-659, Japan; 2Environment and Health Department, Istituto Superiore di Sanitá, 00161 Rome, Italy; alessandro.giuliani@iss.it; 3Faculty of Life and Medical Sciences, Doshisha University, Kyotanabe 610-0394, Japan; keyoshik@mail.doshisha.ac.jp

**Keywords:** genome expression, cell fate decision, self-organized criticality (SOC), critical point, genome engine, genome attractor, biological regulation, transition theory

## Abstract

Elucidation of the genomic mechanism that guides the cell-fate change is one of the fundamental issues of biology. We previously demonstrated that whole genome expression is coordinated by the emergence of a critical point at both the cell-population and single-cell levels through the physical principle of self-organized criticality. In this paper, we further examine the genomic mechanism that determines the cell-fate changes from embryo to cancer development. The state of the critical point, acting as the organizing center of the cell fate, determines whether the genome resides in a super- or sub-critical state. In the super-critical state, a specific stochastic perturbation can spread over the entire system through the “genome engine”, an autonomous critical-control genomic system, whereas in the sub-critical state, the perturbation remains at a local level. The cell-fate changes when the genome becomes super-critical. We provide a consistent framework to develop a time-evolutional transition theory for the biological regulation of the cell-fate change.

## 1. Introduction

A mature mammalian somatic cell can reprogram its state and consequently acquire a very different gene-expression profile through exposure to a few reprogramming stimuli [1]. Such a drastic state change involves the coherent on/off switching of thousands of functionally heterogeneous genes [2]. However, there are fundamental physical difficulties in achieving such large-scale coordinated control on a gene-by-gene basis. These difficulties become more evident in a situation where there is (1) a lack of a sufficient number of molecules to reach a stable thermodynamic state (i.e., breakdown of the central limit theorem) and (2) a consequent stochastic noise due to the low copy numbers of specific gene mRNAs, which induces a substantial instability in genetic product concentrations [3,4].

In our previous studies [5,6,7,8,9,10], we demonstrated that the self-organization of whole-genome expression constitutes a “physically motivated” alternative to gene-specific regulation at both the population and single-cell levels. The mechanism of self-organization, through global genome reprogramming, eliminates the need for physically unfeasible gene-by-gene control of expression.

The core of the self-organization mechanism is the presence of massive system changes elicited by “apparently minor” external causes. To address this problem, Bak and colleagues [11] proposed self-organized criticality (SOC; the Bak–Tang–Wiesenfeld sandpile model). SOC is a general theory of complexity that describes self-organization and emergent order in non-equilibrium systems (thermodynamically open systems). Self-organization occurs at the edge between order and chaos, where the system lies on a “marginally stable” attractor oscillating around an equilibrium position in response to continuous (small) microenvironment perturbations and eventually an “effective” perturbation can spread over the entire system to cause a phase transition [12,13,14,15,16].

SOC builds upon the fact that stochastic perturbations in the great majority of cases propagate locally (i.e., sub-critical state, small avalanches in the sandpile metaphor [11]); however, due to the particularity of the disturbance, the perturbation can spread over the entire system in a highly cooperative manner (i.e., super-critical state, domino effect). As the system approaches its critical point, global behavior emerges in a self-organized manner. The coordinated character of the process stems from the so-called “domino effect” present in all biological signaling processes (e.g., for an allosteric effect, see [17]) where microscopic local effects generalize to the entire system, spreading along “preferential pathways”.

The above-depicted classical concept of SOC has been extended to the cell-fate decision (critical-like self-organization or rapid SOC) through the extension of minimalist models of cellular behavior [18]. The cell-fate decision-making model considers gene regulatory networks to adopt an exploratory process, where diverse the cell-fate options are generated by the priming of various transcriptional programs. In this manuscript, “fate” refers to any stable phenotypic condition of the system. As a result, a cell-fate gene module is selectively amplified as the network system approaches a critical state. Such amplification corresponds to the emergence of the long-range activation/deactivation of genes across the entire genome.

We have adapted the SOC paradigm for the cell-fate decision and investigated whole-genome expression and its dynamics to address the following fundamental questions:-Is there any underlying principle that self-regulates the time evolution of whole-genome expression?-Can we identify a peculiar genome region that guides the super-critical genome and determines the cell-fate change?-Can we delineate a universal mechanism to understand the processes of the cell-fate change, and to further comprehend when and how it occurs?

Our previous studies (see details in [7]) from embryo to cancer development have demonstrated that whole-genome expression is dynamically self-organized through the emergence of a critical point, with the co-existence of three distinct response domains (critical states). Furthermore, dynamic pictures of between-state flux provide a potential universal mechanism of self-organization interpreted in terms of a “genome engine”. An autonomous critical-control genomic system is developed through the highly coherent behavior of low-variance genes (sub-critical state), which in turn, generates a dominant cyclic expression flux with high-variance genes (super-critical state) through the nuclear environment of the cell. This is evident given that gene expression is sorted and grouped according to the temporal variance of expression (*nrmsf*: normalized root mean square fluctuation), which acts as an order parameter for the self-organization of whole-genome expression [6,7]. On the contrary, randomly shuffled gene expression exhibits no evidence of cooperative behavior (no emergence of coordinated activity).

In this report, we further investigate time-series of whole-genome expression data for the cell-fate change: cell differentiation (heregulin (HRG)-stimulated MCF-7 human breast cancer cells compared with non-differentiated epidermal growth factor (EGF)-stimulated MCF-7 cells [19,20]; all-trans retinoic acid (atRA)- and dimethyl sulfoxide (DMSO)-stimulated HL-60 human leukemia cells differentiated to neutrophil cells [21]; differentiation of T helper 17 (Th17) cells from naïve T helper (Th0) cells [22]) and reprogramming (mouse and human early embryo development, [23] and [24], respectively). It is worth noting that these cases must be considered examples of a general mechanism and have been selected based on the availability of a sufficient number of experimental time points for the *nrmsf* value.

We update our previous findings by elucidating both the specific genome region (i.e., critical point, CP) for critical transition and an underlying mechanism that controls the cell-fate change. Our report is organized as follows (Figure 1):

(1) The CP corresponds to the center of mass (CM(*t*_j_): the average value of whole-genome expression at *t* = *t*_j_) (Section 2.1). (2) The dynamics of the center of mass of any stochastic expression converge to that of whole-genome expression (i.e., CM(*t*_j_)). This shows that the CP is the genome attractor (Section 2.2). (3) The switching singular behaviors of the CP transform the genome into a super-critical state (i.e., super-critical genome). This in turn induces a “global expression avalanche”, which is revealed by (4) the probability density function (PDF) of whole-genome expression. (5) Lastly, a cell-fate change occurs after the genome becomes a “super-critical genome” (Section 2.3, Section 2.4, Section 2.5 and Section 2.6) and after the genome passes over a stable point (non-equilibrium fixed point) of the thermodynamically open system. This passing indicates symmetry-breaking which induces coherent perturbation of the genome engine (Section 3.1).

These five points constitute a framework for developing a time-evolutional transition theory of biological regulation.

## 2. Results

### 2.1. Fixed Critical Point: A Specific Group of Genes Corresponding to the Center of Mass of Whole-Genome Expression

The existence of a critical point (CP) is essential for determining distinct response domains (critical states) [7]. To generate a unified model of biological regulation, we must go in depth into specific features of the CP in terms of “sandpile criticality” (Figure 2A). Sandpile criticality emerges when whole-genome expression is sorted and grouped according to the fold-change in expression between two different time points (e.g., between *t* = 0 and *t* = 10 min). For the same groupings, the *nrmsf* value (Section 4.2.1) of the CP in the HRG-stimulated MCF-7 cancer cells (population level) can be estimated (*ln*<*nrmsf*> ~ −2.5: Figure 2B).

Our study of HRG-stimulated MCF-7 cancer cells demonstrated that the temporal group correlation (between-group correlation) along the order parameter (*nrmsf*) reveals a focal point (FP) (see Figure 5B in [6]) when we consider the center of mass (CM) of whole-genome expression (changing in time) as a reference expression point. The grouping (baseline as CM(*t*_j_)) according to the degree of *nrmsf* is called CM grouping,$\;c^{k}\left(t_{j}\right)$ (*k*^th^ group; *k* = 1,2,…, *K*; see Figure 1), where grouping from the CM is distinguished from that of non-reference, $\varepsilon^{k}\left(t\right)$.

Notably, as shown in Figure 2C, the CP corresponds to the zero-expression point in the CM grouping, which explains why the CP is a specific set of genes corresponding to the CM(*t_j_*) and the CP exists as an almost-fixed point in terms of the order parameter *nrmsf*. This feature holds for both single-cell and population data (Figure 2D–I: refer to the natural log of *nrmsf* value for different biological regulations). Therefore, we can develop correlation metrics based on the CM grouping (called CM correlation: Section 4.2.2) to grasp how whole-genome expression can be self-organized through the critical/singular behavior of the CP.

Zimatore et al. [10] demonstrated that CP gene expression (Appendix A) corresponds to attractor-specific values with almost no influence of fluctuation modes, consistent with their zero-expression point in CM grouping.

### 2.2. Critical Point Acting as the Genome Attractor: Mechanism for Genome-Wide Avalanche

Genome expression exhibits coherent-stochastic behavior (CSB) in which coherent behavior emerges from an ensemble of stochastic expression [5,6,7,8]. In CSB mode, gene expression is inherently stochastic but its dynamics follow the center of mass (CM) of expression. Interestingly, CSB is evident at both the whole-genome expression level (Figure 3) and at a specific critical-state level (Figure 3b, Figure 3D in [8]).

To give a proof-of-concept of CSB, we performed a bootstrap simulation to capture two basic signatures of CSB:(1)The stochastic behavior of gene expression shows (relatively low) correlation convergence for randomly selected gene ensembles as the number of elements (*n*) is increased (Figure 3A).(2)The CM of randomly selected gene ensembles (with hundreds of repetitions for the gene ensembles) must dynamically converge to that of whole-genome expression as the number of elements (*n*) is increased (Figure 3B).

This happens at both the population and single-cell levels (Figure 3C,D). The existence of a threshold *n* at around 50 randomly picked genes [6,25] allows us to reproduce CSB with a random choice of *N* genes with *N* > *n*, which is further proof of the reliability of CSB. The constancy of the minimum number of randomly selected genes to reach convergence (Figure 3) is remarkable and suggests the presence of a sort of “percolation threshold” reminiscent of the size of the genetic networks operating in the system.

This convergence clearly reveals that the dynamics of the CM of genome expression describe an attractor of the dynamics of stochastic expression (see also Section 2.6). Therefore, as shown in Figure 3, the CP, the CM according to the degree of *nrmsf*, acts as the genome attractor. Therefore, a change in the CP provokes a genome-wide avalanche over the entire genome expression. Whole-genome expression follows the change in the CP, and this is the origin of coherent gene expression behaviors [26].

Here, it is important to note that CSB does not stem from the law of large numbers under an equilibrium condition, but rather is deduced from the general specific property of an emergent property of an open (non-equilibrium) system.

### 2.3. ON/OFF State of the CP: Cell Fate-Guiding and Global Avalanches

#### 2.3.1. MCF-7 Breast Cancer Development (Cell-Population Level): ON–OFF State of the CP Revealed

We consider the cell-fate change equivalent to a genome-state change (refer to biological discussions in [7]). The genome-state change occurs in such a way that the initial-state SOC control of whole-genome expression, which is in charge of maintaining a largely invariant global gene-expression profile, is destroyed through the erasure of initial-state sandpile criticality [7,9]. Given that the CP acts as a genome attractor, it is essential to understand how changes in the CP lead to the super-critical genome. To elucidate the timing of the change in the CP, we focus on CM correlation dynamics (see Section 4.2.2), i.e., the CM correlation between the initial and other time points: ${\hat{c}}^{k}\left(t_{0}\right).{\hat{c}}^{k}\left(t_{j}\right)$ (*k* = 1,2, …, *K*) over the experimental point, *t_j_*, where ${\hat{c}}^{k}\left(t_{j}\right)$ is the unit vector (unit length) of the *k*^th^ group vector, $c^{k}\left(t_{j}\right)$. CM grouping is used to examine singular behavior at the CP.

In MCF-7 breast cancer cells, activation of the ErbB receptor by HRG and EGF elicits two different biological responses (Figure 4). HRG stimulation induces cell differentiation, whereas EGF stimulation provokes cell proliferation [19,20]. The temporal CM correlation in HRG-stimulated MCF-7 cells reveals a divergent behavior at *t*_j_ = 15 min (Figure 4A, left panel), whereas EGF-stimulated MCF-7 cells (right panel) do not show any divergent behavior. Both responses exhibit sandpile CPs (Figure 4D). These results suggest that the CP possesses both activated and inactivated states, i.e., ON/OFF expression states for a set of genes (critical gene set) corresponding to the CP. In EGF-stimulated MCF-7 cells, the CP is in the inactivated (OFF mode) state, whereas in HRG-stimulated MCF-7 cells, the CP is ON at 10–15 min and thereafter turns OFF.

Direct evidence of the ON/OFF state of the CP is as follows:(1)In Figure 4B, for HRG stimulation, the switching of singular behaviors at 15–20 min occurs at the CP. At the boundary of the CP (*ln*<*nrmsf*> ~ −2.5), the singular behavior exhibits bimodal behavior in the fold-change on the CM grouping. At 15 min, a dominant positive (i.e., fold-change greater than one) singular behavior (*ln*<*nrmsf*> > −2.5) suggests that the swollen coil state of DNA occurs at the CP, while a negative singular behavior (*ln*<*nrmsf*> < −2.5) suggests the compact coil state of DNA. At 20 min, this bimodal behavior switches to a dominant negative singular behavior (> −2.5) with a positive singular behavior (< −2.5). With regard to EGF stimulation, such a switching transition does not occur during the early time points. Note: A negative fold-change occurs given that the reference point is the CM of whole-genome expression. Subtraction of the CM(*t_j_*) from each gene expression gives negative expression.(2)In Figure 4C, the probability density function (PDF) of whole- genome expression [27] shows that at 15 min, around the CP region, the maximum probability density occurs in a positive area, whereas it becomes negative at 15–20 min. This validates that the CP is in the ON state at 15 min and in the OFF state at 20 min. The PDF clearly shows that ON–OFF switching of the CP induces a global avalanche in whole- genome expression.

The fold-changes within expression groups <***c****^k^*(*t_j+1_*)>/<***c****^k^*(*t_j_*)> exhibit a clear first-order phase transition involving genome-sized DNA molecules (see more in Section 3.2). Through our studies, it became evident that the transition occurs as coherent behavior (at a mega-bp scale on the chromosome) and emerges from stochastic expression (coherent-stochastic behavior [8]). In other words, the averaging behavior (mean-field) of group expression corresponds to a coherent transition. In contrast, the ensemble average of the fold-change in individual expressions between two temporal groups, <***c****^k^*(*t_j+1_*)/***c****^k^*(*t_j_*)>, does not reveal such characteristics in the transition. Interestingly, the ensemble average of the time difference in the expression group <***c****^k^*(*t_j+1_*) - ***c****^k^*(*t_j_*)> supports the coherent scenario. This indicates that fluctuation (noise) of coherent dynamics is eliminated (see attached Appendix A).

Regarding the cell-fate change in HRG-stimulated MCF-7 cells, the erasure of initial sandpile criticality occurs after 2 h (Figure 4D). While the cell fate-guiding critical transition occurs at an earlier time point (15–20 min), the cell-fate change happens after 2 h. This time lag is needed to develop a new cell-fate attractor through coordinated local chromatin interaction (see details in [10]).

#### 2.3.2. HL-60 Breast Cancer Cell Development (Cell-Population Level): Timing of Cell-Fate Change

Cell development in HL-60 human leukemia cells further supports the following scenario: (1) the state of the CP changes, such as from an inactivated to an activated state (OFF–ON) or vice versa (ON–OFF), while (2) the switching of singular behaviors occurs and (3) induces a cell fate-guiding global avalanche.

As for the switching behaviors at the CP,

(1)Under atRA stimulation (Figure 5B), the dominant negative fold-change (compact globule state) indicates the OFF state of the CP at 18–24 h, while the dominant positive fold-change (swollen coil state) indicates the ON state at 24–48 h: the change from OFF to ON occurs at the CP. The global avalanche is shown by swelling of the probability density from 18–24 h to 24–48 h and contraction at 48–72 h (Figure 5C).(2)Under DMSO stimulation (Figure 6B), the dominant positive-to-negative fold-change (the opposite of the case with atRA stimulation) occurs from 8–12 h to 12–18 h, i.e., the change from OFF to ON occurs at the CP. At 18–24 h, the CP is neither ON nor OFF. These points are confirmed by the temporal change in the PDF profile (Figure 6C) around the CP: the fold-change (*ln*<*nrmsf*> ~ −2.5) goes from positive to negative to neutral.

Regarding the cell-fate change, the timing of the state-change of the CP through the switching of singular behaviors at the CP coincides with the timing of the erasure of the initial-state sandpile: at 24–48 h for atRA stimulation (Figure 5D) and at 12–18 h for DMSO stimulation (Figure 6D), suggesting that the cell-fate change occurs at 24–48 h for atRA and at 12–18 h for DMSO. The divergent behavior of the temporal CM correlation points to the occurrence of the cell-fate change: the change after the divergent behavior for atRA (Figure 5A) and the change before it for DMSO (Figure 6A). 

As for the DMSO response, it is interesting to observe a multi-step process of erasure of initial-state sandpile criticality (Figure 6D): erasure at 8–12 h; recovery of criticality at 12–18 h; and then erasure again at 18–24 h. Multiple erasures suggest that the cell-population response passes over two SOC landscapes [7] at 8–12 h and 18–24 h.

The results obtained in cancer cells suggest that activation-deactivation of the critical gene set (CP) as the genome attractor plays an important role in the cell-fate change.

### 2.4. Cell-Fate Change in Single-Cell Dynamics

#### 2.4.1. Human Embryo Development: Genome Avalanche Along Low-Expressed Genes

Embryo development starts from a single cell (zygote) and thus corresponds to a completely different situation than in the cases discussed above, where the data refer to averages over millions of cells in a plate (i.e., cell-population level). The CP acts as a genome attractor at the single-cell level and in cell populations (Figure 3). In temporal CM correlation, the CP is a point with no differential (Figure 7A, Figure 8A and Figure 9A), while it appears as a divergent point in cell populations. This feature reveals distinct response domains (critical states) in single-cell genome expression.

In a reprogramming event, the temporal CM correlation for the CP traverses a value of zero (corresponding to random-like behavior) at the 8-cell state for a human embryo cell (Figure 7A). This implies that, after the 8-cell state, the human embryo cell completely erases the initial zygote criticality. Furthermore, this coincides with the erasure of the sandpile-type zygote CP (Figure 7D). In biological terms, this corresponds to erasure of the initial stage of embryogenesis (driven by maternal heredity; see “SOC Control in Human and Mouse Related to the Developmental Oocyte-to-Embryo Transition” in the Discussion [7]).

It is important to note that groups of low-*nrmsf* presenting flattened CM correlations in time (*ln*<*nrmsf>*
**<** −8.0) do not point to a no-response situation; on the contrary, they behave in a highly coherent manner to generate the autonomous SOC mechanism (see, e.g., Figure 6 in [6]).

The switching transition of the CP (Figure 7B) occurs at every cell state change from zygote to blastocyst (only results from the 4-cell state to blastocyst are shown). This suggests that in early embryogenesis, changes in the cell state involve a global change (genome avalanche) in whole-genome expression. Notably, the temporal change in the PDF profile (Figure 7C) shows that a genome avalanche occurs along low-expressed genes (around zero on the *y*-axis), shown as a traveling density wave (higher to lower *nrmsf*) from the 8-cell state to the morula state. The reverse traveling wave (lower to higher *nrmsf*) occurs after the morula state. This traveling wave points to the important role of collective behaviors that emerge in low-expressed genes (refer to local sub-critical state as the generator of the genome engine in Section 3.1). It is intriguing to observe that scaling behaviors (in a log–log plot) emerge, and the most vivid scaling behaviors develop at the 8-cell–morula state.

Therefore, with the results regarding temporal CM correlation, we conclude that a major the cell-fate change (embryonic reprogramming) occurs after the 8-cell state. This coincides with the timing of inverse coherent perturbation of the genome engine (cyclic flux flow), where the genome system passes over the non-equilibrium fixed point after the 8-cell state (the onset of symmetry-breaking occurs; see Section 3.1).

#### 2.4.2. Mouse Embryo Development: Switching Scaling Behaviors in the Cell-Fate Change

In mouse embryo development, the scenario of the cell-fate change described above is confirmed:(1)Complete erasure of the memory of the zygote CP occurs after the late 2-cell state (Figure 8A), where the switching singular transition occurs in the cell-state change from the middle 2-cell state (2-cell M) to the 8-cell state (Figure 8B), showing a major change in the genome.(2)This is confirmed by a temporal change in the PDF (Figure 8C): during cell development from the middle 2-cell to 4-cell states through the late 2-cell state (2-cell L), the whole density profile reflects along the axis of zero-fold change. This clearly manifests as the ON state of the CP at the middle 2-cell – late 2-cell state, and as the OFF state at the late 2-cell – 8-cell state. There are also three distinct scaling behaviors, as in human embryo development. Notably, during the middle 2-cell to 4-cell state through the late 2-cell state, whole-genome expression shifts from up- to downregulation, i.e., marking the occurrence of a global avalanche, and furthermore, there is a reflection of the scaling behavior in the local sub-critical state (*ln*(*nrmsf*) < −11) along the zero-fold change axis (*y* = 0). This reflection illustrates that a coherent switching of folding and unfolding chromatin dynamics occurs, which supports the important role in the local sub-critical state in early embryo development [8].(3)The timing of the cell-fate change is further confirmed by the erasure of zygote sandpile criticality after the late 2-cell state.

#### 2.4.3. Differentiation of Th17 Immune Cells from Th0 Cells: Partial Avalanche Guides the Cell-Fate Change

The cell-fate change in Th17 immune single-cell development again follows the same general scenario. Compared with embryo development, there are several distinct differences worth noting:(1)The CP does not pass over zero temporal correlation with the initial state (*t* = 0 h; Figure 9A), although the correlation response increases in time, which indicates that cell differentiation induces a partial-scale (specific set) change in whole-genome expression (vs. whole genome-scale change in embryonic reprogramming).(2)The switching singular transition at the CP (genome attractor) occurs sequentially from 3 h to 12 h (Figure 9B). This manifests as the OFF state of the CP at 6–9 h and as the ON state at 9–12 h. Figure 9C illustrates that the change in expression for switching is still on a large scale, which is confirmed by the temporal change in the PDF of whole-genome expression. However, there is no distinct scaling behavior as in embryo development.

The timing of the cell-fate change from Th0 to Th17 cells, as in other biological regulations, is determined by the erasure of initial-state sandpile criticality (Figure 9D). This appears at 6 h, where inversion of the singular behaviors of the CP takes place, which is further confirmed by the timing of the inversion of perturbation of the genome engine (see Section 3.1). The OFF–ON switching at the CP guides the cell-fate change.

### 2.5. Systematic Determination of Local Critical States in Genome Expression

We demonstrate that the CP is a fixed point relative to a given biological regulation. Next, based on this fact, we show that critical states in genome expression can be determined systematically for both single-cell and cell population genome expression:(1)Single-cell level: Temporal CM correlation (Figure 7A, Figure 8A and Figure 9A) manifests distinct response domains according to *nrmsf*: low-variance expression (sub-critical state) for the region of flattened correlation, intermediate variance for the near-critical state from the edge of flattened correlation to the CP, and high-variance expression for the super-critical state above the CP (Table 1).(2)Cell-population level: the Euclidean distance (from the highest *nrmsf* group) between the temporal responses of CM grouping (Figure 2: *t* = 0 vs. other experimental time points) reveals critical states (Figure 10; summary: Table 2), where the CP exists at the boundary between near-and sub-critical states (vs. between super-and sub-critical states in a single cell). This occurs for both MCF-7 and HL-60 cancer cell populations. In our previous studies on microarray data (cell-population level), critical states were determined by a change in expression profile by means of Sarle’s bimodality coefficient (Figure 10A; see Figure 9 for MCF-7 and HL-60 cells in [7]). This was accomplished by adding evidence of distinct response domains (super-, near- and sub-critical domains according to *nrmsf*).

### 2.6. Genome Engine Mechanism for SOC-Control of Genome Expression: Heteroclinic Critical-State Attractor System

The existence of distinct critical states at both the single-cell and cell-population levels (Figure 7, Figure 8, Figure 9 and Figure 10) is further demonstrated through the different convergence behaviors of stochastic expression, where the center of mass (CM) of a randomly selected expression of a critical state converges to the CM of the critical state (Figure 11). This further explains the co-existence of distinct local attractors within the genome attractor (Figure 3), i.e., the formation of a heteroclinic critical-state attractor system. The co-existence of distinct local critical states points to the existence of an autonomous critical-control genomic system, i.e., the genome engine. Distinct coherent dynamics in a local critical state emerge from stochastic expression (refer to Figure 10 in [7]) due to the critical-state attractor self-organizing the dynamics of stochastic expression within the local critical state.

Thus, dynamic expression flux analysis (Section 4.2.4) for the CM of critical states can be applied to reveal the genome engine mechanism for describing how autonomous SOC control of genome expression arises. Figure 12 shows that the sub-critical state acts as an internal source of expression flux and the super-critical state acts as a sink. The cyclic flux forms a dominant flux flow that generates a strong coupling between the super- and sub-critical states accompanied by their anti-phase expression dynamics (Figure 13). This results in a change in oscillatory feedback to sustain autonomous SOC control of overall gene expression. The average between-state flux (Figure 12) represents a stable manifold for the thermodynamically open system. Formation of the dominant cyclic flux provides a genome engine metaphor for SOC control mechanisms pointing to a universal mechanism in the gene-expression regulation of mammalian cells for both the population and single-cell levels. As demonstrated in Section 3.1, global perturbation stems from the change in CP, which either enhances or suppresses the genome engine, and thus transforms the cell-fate change.

## 3. Discussion and Conclusion

### 3.1. Genomic Dynamics Determining the Cell-Fate Change from Embryo to Cancer: Cell-Fate Change Passing through a Non-Equilibrium Fixed Point

The different examples of cell-fate determination (cell differentiation and reprogramming in embryo development) have highlighted two crucial common elements: (1) the critical point (CP) acts as the organizing center of the cell fate, and (2) a sort of genome engine (Figure 12) fuels the transition and constitutes the basic mechanism of the cell-fate change. This genome engine works by canalizing the stochastic expression of local critical states into coherent behavior, provoking the motion of the center of mass (CM) of genome expression (i.e., change in the genome attractor). This is exactly what happens when an artificial engine transforms combustion (or any other energy source) into motion.

The CM acts as an attractor at both the whole-genome (Figure 2) and critical-state levels (Figure 11). Based on this fact, the expression flux (effective force) approach (Section 4.2.4) was developed to reveal dynamic interaction flux between critical-state attractors. Figure 13 demonstrates that the expression flux approach is instrumental for revealing a heteroclinic critical transition at the occurrence of a global avalanche that guides the coherent behaviors that emerge in the critical states. Interaction flux of between-state attractors is the underlying basic mechanism of epigenetic self-regulation, incorporating a rich variety of transcriptional factors and non-coding RNA regulation to determine the coherent oscillatory behaviors of critical states.

The flux dynamics approach has been further developed to analyze the quantitative evaluation of the degree of non-harmonicity and time-reversal symmetry-breaking in nonlinear coupled oscillator systems [28].

Thus, we summarize how and when the cell-fate change occurs as follows:

(1) How the cell-fate change occurs:

The intersection of interaction fluxes (Figure 14 for cell population and Figure 15 for single cell; see the dashed ovals in interaction expression flux) occurs just before a cell-fate change. This intersection means that, thermodynamically, the genomic system lies at a non-equilibrium fixed point (e.g., see Figure 14C: atRA stimulation at 24 h with small interaction fluxes; see also Section 4.2.4), which suggests that before the cell-fate change, time-reversal symmetry is broken through passing over a non-equilibrium fixed condition. In terms of the genome engine, around the cell-fate change, a global perturbation induces either enhancement or suppression of the genome engine, where there is a dominant cyclic flux flow between the super- and sub-critical states (Table 3: single cell; Table 4: cell population). In HL-60 cells (cell population), the genome engine is enhanced before the cell-fate change and suppressed (enhancement-suppression) thereafter. On the contrary, a reverse process of suppression–enhancement takes place in MCF-7 cancer cells (Figure 14A; see more in [7]). In single-cell cases (embryo development and Th17 immune cells), a suppression–enhancement of the genome engine occurs (Figure 15). The varying sequences of perturbation of the genome engine may stem from different stages of the suppressive pressure on cell differentiation against cell proliferation.

(2) When the cell-fate change occurs:

Cancer cell differentiation is accompanied by a change from a compact globule (OFF) to a swollen coil (ON) or vice versa at the CP (see Figure 4, Figure 5 and Figure 6). As shown in Figure 5C and Figure 6C (HL-60 cells), a global genome avalanche occurs at the timing of the cell-fate change. For MCF-7 cells (HRG stimulation), the cell-fate change (after 2 h: Figure 4D) occurs after the genome avalanche (see details of the attractor mechanism in [10]). The global impact provoked by the CP is further supported by the fact that (1) EGF-stimulated MCF-7 cells, where the CP is OFF with no cell differentiation, induce only local perturbation (Figure 16C) and (2) HL-60 cells coincide with the timing of global perturbation (Figure 16D).

Regarding the cell-fate change in single-cell dynamics, a global perturbation (Figure 16A,B) occurs for the genome engine (after the late 2-cell and 8-cell stages and after 6 h for mouse, human embryo, and Th17 cells, respectively), where the timing coincides with that of the cell-fate change (i.e., erasure of the initial-state CP memory). This suggests that the cell fate-guiding change in the CP corresponds to the erasure of initial-state sandpile criticality. For embryo reprogramming, this picture is supported by the fact that temporal CM correlation of the CP from the initial state (zygote) passes zero-correlation (i.e., erasure of the initial-state CP memory: Figure 7A and Figure 8A). Due to the genome attractor, the change in the CP (the edge of criticality) has a global impact on the entire genome expression through perturbation of the genome engine: global perturbation leads to a genome avalanche at the timing of the cell-fate change (Figure 7C, Figure 8C and Figure 9C; also refer to Figure 13 and Biological Discussion in [7]).

Therefore, elucidation of the activation-deactivation mechanism for the CP (involving more than hundreds of genes) through a coordinating change in chromatin structure [10] uncovers how the genome engine is either enhanced or suppressed around the timing of the cell-fate change. Furthermore, our SOC hypothesis is expected to predict when and how the cell-fate change occurs in different situations (e.g., cancer, induced pluripotent stem cells (iPS cells), etc.). The universality of the proposed model does not stem from our experimental results, but rather from the existence of very basic physical constraints (e.g., chromatin dimension and general organization, and phase transition phenomenology) independent of biological specificities.

### 3.2. Synergetic Behaviors of Mega Base Pair-Size DNAs Through a Phase Transition

Up to this point, we have focused on the genomic dynamics that convert the signatures of the cell-fate change into self-organized complexity. Here, we should recall that our first objective was to recognize the unfeasibility of precise gene-by-gene regulation in the presence of the extremely compact folding of a 2-meter-long molecule that is compressed within a few microns of space. This calls for a search for a structural counterpart of the above-discussed model. In this respect, it is worth noting that the CP shows a clear bimodal expression distribution reminiscent of the swollen coil (ON) and compact globule (OFF) DNA transition, corresponding to inversion of the intra-chain phase segregation of the coil and globule states. This behavior closely resembles the intrinsic characteristics of the first-order phase transition inherent in genome-sized DNA molecules. The activation-deactivation of the CP is essential for the occurrence of the cell-fate change.

As for the phase transition of mega base pair-size DNA, until the late 20th century, it was believed that single polymer chains, including DNA molecules, always exhibited a cooperative but mild transition between the elongated coil and compact globule states, which was considered neither a first-order nor a second-order phase transition [30,31]. More recently, it has become evident that long DNA molecules above the size of several tens of kilo base pairs exhibit characteristics that help them undergo a large discrete transition, i.e., first-order properties related to the coil–globule transition [32,33,34]. Such first-order characteristics are rather general for semi-flexible polymers, especially polyelectrolyte chains such as giant DNA molecules. Additionally, it is important to note that insufficient charge neutralization in the globule state causes instability and leads to the generation of intra-chain segregation of individual single genomic DNA [35,36] (see, e.g., Figure 8B). When such instability is enhanced with long-range Coulombic interaction, the characteristic correlation length tends to become shorter, corresponding to generation of the critical state in the transition. Such transitional behavior that accompanies the folding transition of DNA is also observed for reconstructed chromatin [37,38,39].

Thus, the state change in the CP is suggested to guide the synergistic effect on mega base pair-size DNAs through the phase transition in the cell-fate change. Our recent study based on non-linear signal analysis techniques (recurrence quantification analysis and principal component analysis) revealed that chromatin remodeling played a role as the material basis of SOC-controlled genome expression [10]. A genome-wide expression avalanche stems from the coordinated activity of positional-based (not biological-functional) local chromatin interaction. The cell fate-guiding critical transition occurs based on the non-linear resonance of two major expression fluctuation modes, which establish an autonomous genome engine through activation of the CP. Elucidation of the link between the spatial position on the chromosome and co-regulation together with the identification of specific locations on the genome devoted to the generalization of perturbation stimuli provides a molecular basis for the self-organization dynamics of genome expression and the cell-fate decision.

### 3.3. CP Potentially Acting as the Center of Genome Computing

The singular behavior at the CP exhibits bimodal folding and unfolding chromatin dynamics through synergistic or cooperative behavior in higher-order structural transition of genomic DNA, which suggests the existence of a complex network of mega-sized ON/OFF DNA phase transitions in the cell-fate change. This interpretation was recently confirmed by the evidence of a massive re-arrangement of peri-centromeric nuclear regions correspondent to gene expression transitions [40]. The necessary link between chromatin remodeling and gene expression regulation was experimentally assessed [41]. These further imply that the CP may act as the center of genome computing, where chromatin remodeling is the material basis of such computation. 

The time-dependent behavior of the genomic DNA transition follows a kinetic equation to exhibit cubic nonlinearity. A simultaneous change in the translational and conformational entropy of giant DNA together with surrounding counter ions causes bimodality in the free energy, which in turn speeds the transition under cubic nonlinearity in the kinetic equation [42,43]. The incorporation of negative feedback in cubic nonlinearity leads to fundamental characteristic nerve firings. Therefore, the occurrence of SOC in the brain’s complex neural networks suggests that the state change in the CP is a computing process through coordinated DNA structural transitions to guide the cell-fate change. Elucidation of the computing process at the CP could reveal the control mechanism for the cell-fate change in a desirable manner, i.e., the existence of “genome intelligence”. Here, note that cubic nonlinearity in the time-differentiation equation corresponds to a kinetic representation of the bimodal free energy, and criticality is generally interpreted based on the symmetry argument under this type of energy landscape [44].

### 3.4. Biological Meaning of the CP

As noted above, we investigated a new interpretation of the specificity of biological regulation without relying on an improbable “Maxwell’s demon” ability to select the “right genes” interspersed in a two-meter long molecule constrained within the few cubic microns of the cell nucleus. The genes exert specific biological functions and an analysis of such functions could give us further hints regarding the mechanism of the cell-fate change. The strict link between chromatin remodeling and the phase transition is in fact mirrored by the gene composition of the super-critical (genome) state. Appendix A based on principal component analysis shows the 200 genes that are most strongly affected by the second principal component (PC2) of gene expression variance in HRG-treated MCF7 cells, as explained in [10]. PC2 corresponds to the super-critical set referring to the component with genes that show higher temporal variation. The continuous fluctuation in time of this component (orthogonal by construction to the first principal component (PC1), which corresponds to the equilibrium position, i.e., the attractor) allows for both continuous adaptation to the changing micro-environment (small avalanches) and fueling of the state change (phase transition). The biological role of these genes (endowed with statistically significant z-scores on PC2 and thus being the “top players” in the supercritical state, see Appendix A) confirms that chromatin remodeling is the basic structural driver of the genome engine. The bolded genes in Appendix A are all consistent with structural chromatin remodeling obtained by both histone modifications and DNA breaks that allow for global nuclear re-organization. This nuclear re-organization requires a cross-talk with the perinuclear environment (note the prevalence of actin, myosin, cadherin and integrin genes) and activates known signaling hubs such as proto-oncogenes c-FOS and c-Myc, Mitogen-activated protein (MAP) kinase, and transformation-related protein 53 (TP53) (Appendix A). This is a stringent, albeit indirect, proof of the proposed model in terms of mainstream biology.

### 3.5. Final Remarks

We highlighted the basic invariance of the model in systems from embryo development to cancer. The different model cases are considered as examples of a general mechanism, where the need for a relevant time course to be analyzed is the only inclusion criterion, together with the presence of a relevant perturbation of the studied system. We can anticipate that other kinds of cells will behave according to the SOC model with differences linked to the presence (or absence) of a stable phenotype change during the time course considered. As aptly stated in [45], criticality is a general feature of biological regulation.

Finally, we summarize our findings in the following points:(1)The critical point (CP) acts as the center of the cell fate.(2)How the cell-fate change occurs: Before the cell-fate change, the genomic system passes over the stable point (non-equilibrium fixed point) of the thermodynamically open system. The genome engine, an emergent dynamic property of an autonomous interaction flux between local critical states (distinct expression domains in whole-genome expression), is enhanced or suppressed to induce coherent perturbation of the dominant cyclic flux between local super- and sub-critical states.(3)When the cell-fate change occurs: The change occurs at the timing of the erasure of the initial-state sandpile CP. At this time, a global genome avalanche occurs, except in HRG-stimulated MCF-7 cell differentiation, with a time lag between the genome avalanche and the cell-fate change (for further details of this mechanism, see [10]).

The genome engine suggests that the activation-deactivation mechanism of the CP elucidates how global perturbation occurs on self-organization through a change in signaling due to external or internal stimuli. Coordinated chromatin dynamics emerge to guide reprogramming through SOC control. A recent study found that the dynamics of the high-order structure of chromatin exhibit liquid-like behavior [46], which could be a crucial characteristic that enables the genome to exhibit SOC gene expression control for determination of the cell fate.

Further studies on these matters are needed to clarify the underlying fundamental molecular mechanism(s). The development of a theoretical foundation for the autonomous critical control mechanism in genome expression as revealed in our findings is expected to lead to new insights regarding for a general control mechanism for determination of the cell fate and genome computing.

For now, we can safely affirm that the strong interactions among genes with very different expression variance and physiological roles push for a complete reshaping of the current molecular-reductionist view of biological regulation focused on a single “significantly affected” gene to explain these regulation processes. The view that the genome acts as an integrated dynamic system is here to stay.

## 4. Materials and Methods

### 4.1. Biological Data Sets

We analyzed mammalian transcriptome experimental data for seven distinct the cell-fates in different tissues.

#### 4.1.1. Cell population:

1. Microarray data of the activation of ErbB receptor ligands in human breast cancer MCF-7 cells by EGF and HRG; Gene Expression Omnibus (GEO) ID: GSE13009 (*N* = 22,277 mRNAs; experimental details in [29]) at 18 time points: *t*_1_ = 0, *t*_2_ = 10, 15, 20, 30, 45, 60, 90 min, 2, 3, 4, 6, 8, 12, 24, 36, 48 h, *t*_T = 18_ = 72 h.

2. Microarray data of the induction of terminal differentiation in human leukemia HL-60 cells by DMSO and atRA; GEO ID: GSE14500 (*N* = 12,625 mRNAs; details in [21]) at 13 time points: *t*_1_ = 0, *t*_2_ = 2, 4, 8, 12, 18, 24, 48, 72, 96, 120, 144 h, *t*_T=13_ = 168 h.

#### 4.1.2. Single-cell:

3. RNA-Seq data of early embryonic development in human and mouse developmental stages in RPKM (reads per kilobase of transcript, per million mapped reads) values; GEO ID: GSE36552 (human: *N* = 20,286 RNAs) and GSE45719 (mouse: *N* = 22,957 RNAs) with experimental details in [24] and [23], respectively.

We analyzed 7 human and 10 mouse embryonic developmental stages as listed below:

Human: oocyte (*m* = 3), zygote (*m* = 3), 2-cell (*m* = 6), 4-cell (*m* = 12), 8-cell (*m* = 20), morula (*m* = 16), and blastocyst (*m* = 30);

Mouse: zygote (*m* = 4), early 2-cell (*m* = 8), middle 2-cell (*m* = 12), late 2-cell (*m* = 10), 4-cell (*m* = 14), 8-cell (*m* = 28), morula (*m* = 50), early blastocyst (*m* = 43), middle blastocyst (*m* = 60), and late blastocyst (*m* = 30), where *m* is the total number of single cells.

4. RNA-Seq data of T helper 17 (Th17) cell differentiation from mouse naive CD4+ T cells in FPKM (fragments per kilobase of transcript, per million mapped reads) values, where Th17 cells were cultured with anti-IL-4, anti-IFNγ, IL-6 and TGF-β, (details in [22]; GEO ID: GSE40918 (mouse: *N* = 22,281 RNAs) at 9 time points: *t*_1_ = 0, *t*_2_ = 1, 3, 6, 9, 12, 16, 24 h, *t*_T=9_ = 48 h. For each time point, the reference sample numbers are listed: GSM1004869-SL2653 (*t* = 0 h); GSM1004941-SL1851 (*t* = 1 h); GSM1004943-SL1852 (*t* = 3 h); GSM1005002-SL1853 (*t* = 6 h); GSM1005003-SL1854 (*t* = 9 h); GSM1004934-SL1855 (*t* = 12 h); GSM1004935,6,7-SL1856, SL8353, SL8355 (*t* = 16 h; average of three data points); GSM1004942-SL1857 (*t* = 24 h); GSM1004960-SL1858 (t = 48 h).

The colors used in the various plots throughout this report are based on the experimental events and were assigned as follows: black as the initial event, purple as the second event, and subsequent events as blue, dark cyan, dark green, dark yellow, brown, orange, red, dark pink, and pink.

For microarray data, the robust multichip average (RMA) was used to normalize expression data for further background adjustment and to reduce false positives [47,48,49].

For RNA-Seq data, RNAs with RPKM and FPKM values of zero over all of the cell states and experimental time points were excluded. In the analysis of sandpile criticality, random real numbers in the interval [0, *a*] generated from a uniform distribution were added to all expression values (only in Figure 7D, Figure 8D and Figure 9D). This procedure avoids the divergence of zero values in the logarithm. The robust sandpile-type criticality through the grouping of expression was checked by changing a positive constant: *a* (0 < *a* < 10); we set *a* = 0.001. Note: The addition of large random noise (*a* >> 10) destroys the sandpile CP.

### 4.2. Methods

#### 4.2.1. Normalized Root Mean Square Fluctuation (*nrmsf*)

*Nrmsf* (see also Methods in [6]) is defined by dividing *rmsf* (root mean square fluctuation) by the overall maximum {*rmsf_i_*} (Equation (1)):(1)$$rmsf_{i}=\sqrt{\frac{1}{S}\sum_{j=1}^{S}{\left(\varepsilon_{i}\left(s_{j}\right)-\langle \varepsilon_{i}\rangle \right)}^{2}},$$
where *rmsf_i_* is the *rmsf* value of the *i*^th^ RNA expression, which is expressed as *ε_i_*(*s_j_*) at a specific cell state *s_j_* or experimental time (e.g., in mouse embryo development, *S* = 10: *s*_1_ = zygote, early 2-cell, middle 2-cell, late 2-cell, 4-cell, 8-cell, morula, early blastocyst, middle blastocyst, and *s*_10_ = late blastocyst), and $\langle \varepsilon_{i}\rangle$ is its expression average over the number of cell states. Note: *Nrmsf* is a time-independent variable, where *nrmsf* acts as an order parameter of self-organization [6,7].

#### 4.2.2. CM Correlation Analysis

To investigate transition dynamics, the correlation metrics based on the center of mass (CM) grouping, the CM correlation, was built upon the following basic statistical formalization:

(1) CM grouping: genome expression is considered as an *N*-dimensional vector, where the average value (CM) of whole-genome expression at *t* = *t*_j_ is subtracted from each expression (refer to Figure 1). Next, the whole-genome expression is sorted and grouped according to the degree of *nrmsf,* where CM grouping has *K* groups and within each group there are *n* number of expressions (*N* = *K*^.^*n*): *N*-dimensional CM grouping vector,$C\left(t_{j}\right)=\left(c^{1}\left(t_{j}\right),c^{2}\left(t_{j}\right),\ldots ,c^{k}\left(t_{j}\right),\ldots ,c^{K}\left(t_{j}\right)\right)$; $c^{1}\left(t_{j}\right)$ and $c^{K}\left(t_{j}\right)$ are the highest and lowest group vectors of *nrmsf*, respectively. Here, the unit vector of the *k*^th^ vector $c^{k}\left(t_{j}\right)$ is defined as ${\hat{c}}^{k}\left(t_{j}\right)=\frac{c^{k}\left(t_{j}\right)}{\left|c^{k}\left(t_{j}\right)\right|}$. Note that the elements less than *n* in the last group (the lowest *nrmsf*) have been removed from the analysis.

(2) If we keep in mind that correlation corresponds to the cosine of the angle between unit vectors, i.e., the inner product of unit vectors ($\cos\theta =\hat{a}.\hat{b}$, θ:angle; $\hat{a}.\hat{b}$: dot product (scalar) of unit vectors:$\;\hat{a}$,$\;\hat{b}$), two different CM correlations can be considered:

(i) Spatial CM correlation: for a given time point ($t=t_{j}$), the development of CM correlation between the first group (highest *nrmsf* group) and other vectors: ${\hat{c}}^{1}\left(t_{j}\right).{\hat{c}}^{k}\left(t_{j}\right)$; (*k* = 2,3,…*K*).

(ii) Temporal CM correlation: for a given group (*k*), the development of CM correlation between the initial and other experimental points: ${\hat{c}}^{k}\left(t_{1}\right).{\hat{c}}^{k}\left(t_{j}\right)$ (*k* = 1,2,3,…*K*) over experimental time points, *t_j_* (see Biological Data Sets).

#### 4.2.3. Probability Density Function (PDF)

The robustness of gene expression clustering by means of the density analysis of noisy gene-expression profiles was demonstrated by Shu et al. [24]. The probability density function (PDF) based on the Gaussian kernel is examined here. We consider an *N*-dimensional whole-genome expression vector for the natural logarithm (natural log) of expression and its fold-change vector, where the natural log of the whole expression vector is defined as *ln*(*ε*(*t*_j_)) = (*ln*(*ε*_1_(*t*_j_)), *ln*(*ε*_2_(*t*_j_)),…, *ln*(*ε_N_*(*t*_j_))) with the *i*^th^ expression, *ε_i_*(*t*_j_) at *t* = *t*_j_ (Figure 1) and its fold-change vector is defined as *ln*(*ε*(*t_j_*)/*ε*(*t_k_*)) = (*ln*(*ε*_1_(*t_j_*)/*ε*_1_(*t_k_*)), *ln*(*ε*_2_(*t_j_*)/*ε*_2_(*t_k_*)),…, *ln*(*ε_N_*(*t_j_*)/*ε_N_*(*t_k_*)). This satisfies the logarithm of a quotient: *ln*(*ε*(*t_j_*)/*ε*(*t_k_*)) = *ln*(*ε*(*t*_j_)) − *ln*(*ε*(*t_k_*)).

Next, we consider these vectors with regard to their CM (called the CM expression vector), the CM natural log of the whole-genome expression vector and its fold-change vector. The CM natural log of the whole-genome expression vector is defined as *ln*(c(*t*_j_)) = *ln*(*ε*(*t*_j_)) − CM(*ln*(*ε*(*t*_j_))) and CM(*ln*(*ε*(*t*_j_))) = $\frac{1}{N}\sum_{i=1}^{N}ln\left({}_{i}\left(t_{j}\right)\right)$ and its fold-change vector is *ln*(*c*(*t*_j_)/*c*(*t_k_*)) = *ln*(*ε*(*t_j_*)/*ε*(*t_k_*)) − CM(*ln*(*ε*(*t_j_*)/*ε*(*t_k_*))). This vector representation again satisfies the logarithm of a quotient: *ln*(*c*(*t_j_*)/*c*(*t_k_*)) = *ln*(*c*(*t*_j_)) − *ln*(*c*(*t_k_*)).

From embryo to cancer development, global avalanches in changes in the CP are captured, and furthermore, this presents scaling behaviors in addition to the traveling expression wave in human and mouse embryo development (RNA-Seq data) (see Figure 7C and Figure 8C), which suggests coordinated DNA folding-unfolding dynamics.

#### 4.2.4. Expression Flux Analysis: Expression Flux Between Critical-State Attractors

We developed an expression flux analysis to describe the genome engine mechanism for both single-cell and population genome expression (see Figure 12). The key fact is that the dynamics of coherent behavior emerging from stochastic expression in distinct critical states (coherent-stochastic behavior: CSB) follow the dynamics of the CM of a critical state. This convergence shows that the CM of a critical state acts as a critical-state attractor for stochastic expression within the critical state (Figure 11). We developed the expression flux approach to reveal the dynamic interaction flux between critical-state attractors [6,7,8].

The CSB in a critical state corresponds to the scalar dynamics of its CM. The numerical value of a specific critical state (i.e., super-, near- or sub-critical state) is represented by *X*(*s_j_*) at a specific experimental event (*s_j_*), where an experimental event (*s_j_*) corresponds to a cell state or an experimental time point. The expression flux between critical states is interpreted as a non-equilibrium system and evaluated in terms of a dynamic network of effective forces, where interaction flux is driven by effective forces between different critical states and can be described by a second-order time difference. It is important to note that the oscillatory phenomenon is interpreted using a second-order difference equation with a single variable. This is equivalent to the inhibitor-activator dynamics given by a couple of first-order difference equations with two variables.

The flux dynamics approach has been further developed to analyze the quantitative evaluation of the degree of non-harmonicity and time-reversal symmetry-breaking in nonlinear coupled oscillator systems [28].

The basic formulas of expression flux dynamics are given as follows:

Net self-flux of a critical state: The net self-flux, the difference between IN flux and OUT flux, describes the effective force on a critical state. This net self-flux represents the difference between the positive sign for incoming force (net IN self-flux) and the negative sign for outgoing force (net OUT self-flux); the CM from its average over all cell states represents up- (or down-) regulated expression for the corresponding net IN (or OUT) flux.

The effective force is a combination of incoming flux from the past to the present and outgoing flux from the present to the future cell states (Equation (2)):(2)$$\begin{array}{ll} f\left(X\left(s_{j}\right)\right)=\frac{\Delta P}{\Delta s} & =\frac{1}{\Delta s}\left\{\frac{\left(X\left(s_{j}\right)-X\left(s_{j-1}\right)\right)}{\Delta s_{j}}-\frac{\left(X\left(s_{j+1}\right)-X\left(s_{j}\right)\right)}{\Delta s_{j+1}}\right\}-\langle f\left(X\right)\rangle \\ & =\left(\mathrm{IN}\;\mathrm{flux}-\langle \mathrm{IN}\;\mathrm{flux}\rangle \right)-\left(\mathrm{OUT}\;\mathrm{flux}-\langle \mathrm{OUT}\;\mathrm{flux}\rangle \right), \end{array}$$
where Δ*P* is the change in momentum with a unit mass (i.e., the impulse: *F*Δ*s* = Δ*P*) and the natural log of the average (<…>) of a critical state, $X\left(s_{j}\right)=ln\langle \frac{1}{N_{C}}\sum_{i=1}^{N_{C}}\varepsilon_{i}\left(s_{j}\right)\rangle$ with the *i*^th^ expression $\varepsilon_{i}\left(s_{j}\right)$ at the *j*^th^ experimental event, *s* = *s_j_* (*N_C_* = the number of RNAs in a critical state; refer to Table 1 and Table 2); the average of net self-flux over the number of critical states, <*f*(X)> = <INflux> − <OUTflux>.

Here, scaling and critical behaviors occur in log–log plots of group expression, where the natural log of an average value associated with group expression such as *ln*<*nrmsf*> or *ln*<*expression*> is taken. Thus, in defining expression flux, the natural log of the average expression (CM) of a critical state is considered.

It is important to note that each embryo cell state is considered as a statistical event (a statistical event does not necessarily coincide with a biological event) and its development is considered as a time arrow (time-development) when evaluating the average of group expression: the fold-change in expression and temporal expression variance (*nrmsf*). This implies that an interval in the dynamic system (Equation (2)) is evaluated as the difference in events, i.e., Δ*s*_j_ = *s*_j_ − *s*_j−1_= 1 and Δ*s* = *s*_j+1_ − *s*_j−1_ = 2 in embryo development, as well as the difference in experimental times such as in cell differentiation (the actual time difference can be considered as scaling in time). We then evaluate a force-like action in expression flux.

The interaction flux of a critical state: The interaction flux, representing the flux of a critical state *X*(*s_j_*) with respect to another critical state (super-, near-, sub-) or the environment (E: milieu), *Y_j_*, can be defined as (Equation (3)):(3)$$f\left(X\left(s_{j}\right);Y\right)=\frac{1}{\Delta s}\left\{\frac{\left(X\left(s_{j}\right)-Y\left(s_{j-1}\right)\right)}{\Delta s_{j}}-\frac{\left(Y\left(s_{j+1}\right)-X\left(s_{j}\right)\right)}{\Delta s_{j+1}}\right\}-\langle f\left(X;Y\right)\rangle ,$$
where, again, the first and second terms represent IN flux and OUT flux, respectively, and the net value (i.e., IN flux − OUT flux) represents incoming (IN) interaction flux from *Y* for a positive sign and outgoing (OUT) interaction flux to *Y* for a negative sign. *Y* represents the numerical value of either a specific critical state or the environment, where a state represented by *Y* is different from one represented by *X.*

With regard to the global perturbation event, the net kinetic energy flux [7] clearly reveals the timing of the global perturbation event (Figure 16) (Equation (4)):(4)$$K\left(X\left(s_{j}\right)\right)=\frac{1}{2}\left\{{\left(\frac{\left(X\left(s_{j}\right)-X\left(s_{j-1}\right)\right)}{\Delta s_{j}}\right)}^{2}-{\left(\frac{\left(X\left(s_{j+1}\right)-X\left(s_{j}\right)\right)}{\Delta s_{j+1}}\right)}^{2}\right\}-\langle K\left(X\right)\rangle ,$$
where the kinetic energy of the CM for the critical state with unit mass at *s* = *s_j_* is defined as 1/2^.^$\;v\left(s_{j}\right)$^2^ with average velocity: $v\left(s_{j}\right)\;\equiv \;\frac{X\left(s_{j}\right)-X\left(s_{j-1}\right)}{\Delta s_{j}}$.

Net self-flux as summation of interaction fluxes: Due to the law of force, the net self-flux of a critical state is the sum of the interaction fluxes with other critical states and the environment (Equation (5)):(5)$$f\left(X\left(s_{j}\right)\right)=\overset{M=2}{\underset{i=1}{\sum }}f\left(X\left(s_{j}\right);A_{i}\right)+f\left(X\left(s_{j}\right);E\right),\;$$
where state $A_{i}\in \left\{\mathrm{Super},\;\mathrm{Near},\;\mathrm{Sub}\right\}$ with $A_{i}\neq$
*X*, and *M* is the number of internal interactions (*M* = 2), i.e., for a given critical state, there are two internal interactions with other critical states. Equation (5) tells us that the sign of the difference between the net self-flux and the overall contribution from internal critical states,$\;f\left(X\left(s_{j}\right)\right)-\sum_{i=1}^{M=2}f\left(X\left(s\right);A_{i}\right),$ reveals incoming flux (positive) from the environment to a critical state or outgoing flux (negative) from a critical state to the environment. When the difference in all critical states is zero, the genome system itself lies in a stable point (non-equilibrium fixed point) of a thermodynamically open system (no average flux flow from the environment). Average between-state flux (Figure 12) represents a stable manifold of the thermodynamically open system.

##### Order of averaging: $ln\langle \frac{1}{n}\sum_{i=1}^{n}\varepsilon_{i}\left(s_{j}\right)\rangle$ vs. $\frac{1}{n}\sum_{i=1}^{n}\ln\langle \varepsilon_{i}\left(s_{j}\right)\rangle$

Here, we need to address the previous result of expression flux dynamics in mouse single-cell genome expression [8], where the expression of a critical state was taken as $X\left(s_{j}\right)=\frac{1}{N_{C}}\sum_{i=1}^{N_{C}}ln\langle \varepsilon_{i}\left(s_{j}\right)\rangle$, which has a different order of operations: first take the natural log of the expression and then calculate the average. Hence, in flux dynamics, we examine whether or not the mathematical operations of averaging and the natural log, i.e., between $ln\langle \frac{1}{n}\sum_{i=1}^{n}\varepsilon_{i}\left(s_{j}\right)\rangle$ and $\frac{1}{n}\sum_{i=1}^{n}ln\langle \varepsilon_{i}\left(s_{j}\right)\rangle$, can be exchanged (mathematically commuted). In microarray data, flux behaviors do not change much between different orders of actions, whereas in RNA-Seq data, they are not commuted because its data structure has many zero values. While the addition of small random noise to the log of expression, $ln\langle \varepsilon_{i}\left(s_{j}\right)\rangle$ (used for the previous result [8], a noise-sensitive case) has a good effect, the addition of such noise to $ln\langle \frac{1}{n}\sum_{i=1}^{n}\varepsilon_{i}\left(s_{j}\right)\rangle$), used for noise-insensitive cases (such as in this report), does not have a good effect. Therefore, due to this sensitivity, micro-array data may be better for expression flux analysis (see Figure 13). Although the detailed dynamics of interaction flux changes by ordering actions differently for RNA-Seq data (e.g., Figure 6 in [8]), two important characteristics in the genome engine, the formation of dominant cyclic flux between super- and sub-critical states and the generator role of the sub-critical state, do not change (invariant features). Thus, we conclude that the concept of the genome engine is quite robust.

## Figures and Tables

**Figure 1 ijms-21-04581-f001:**
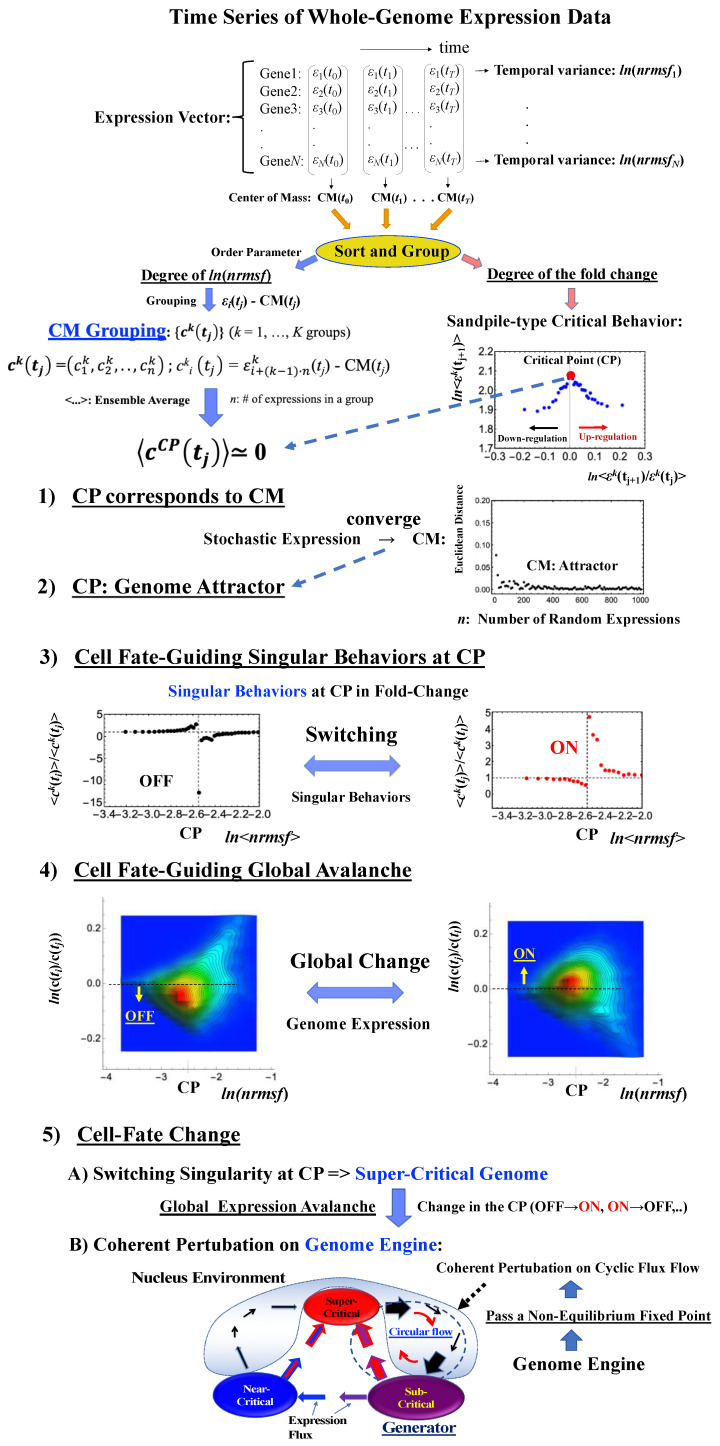
Schematic representation of the genomic mechanism for the cell-fate change. (**1**–**4**) The critical point (CP) corresponds to the center of mass (CM) and represents a specific set of critical genes acting as a genome attractor. The CP has activated (ON) and deactivated (OFF) states. ON/OFF switching of the CP state occurs through switching of its singular behaviors, i.e., change in the critical transition of a specific critical gene set. Changes in the state of the CP, such as ON–OFF or OFF–ON switching, guide the genome into a super-critical state; thus, these perturbations can spread over the entire system in a highly cooperative manner. Chromatin remodeling plays a role as the material basis of the self-organized critical (SOC) control of genome expression [10]. (**5**) Due to the CP acting as a genome attractor, the self-organization of gene expression develops an autonomous critical-control genomic system (genome engine) through the formation of dominant cyclic flux between local critical states (distinct expression domains according to the degree of *nrmsf*: Section 4.2.1), where the local sub-critical state is the generator (see details in [7]). Coherent perturbation of the genome engine through changes in the CP (ON to OFF, OFF to ON, etc.) drives the cell-fate change. Before the cell-fate change, the genome (expression) system passes through a non-equilibrium fixed point (stable point of the thermodynamically open system). These five points support the development of a time-evolutional transition theory of biological regulation. Throughout this report, the whole-genome expression vector **c**(*t*_j_) at *t* = *t*_j_ represents the CM expression vector, in which the center of mass of whole-genome expression, CM(*t*_j_) is subtracted from each expression (Section 4.2.3).

**Figure 2 ijms-21-04581-f002:**
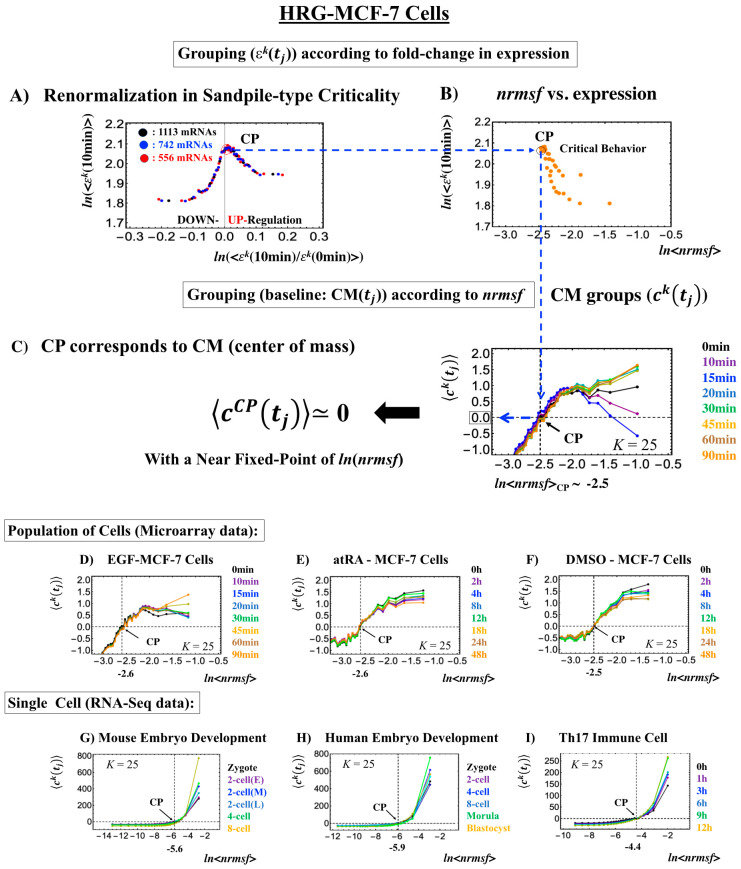
Fixed CP corresponding to the center of mass (CM) of genome expression: (**A**) Different numbers of groups ($\varepsilon^{k}\left(t\right)$, *k* = 1,2,…,*K*: non-CM grouping; black: *K* = 20; blue: 30; red: 40 groups; groups contained 1113, 742, and 556 mRNA species, respectively) in HRG-stimulated MCF-7 cells followed the same sandpile-type critical behavior. This reveals the existence of scaling behavior (i.e., renormalization). (**B**) *ln*<*nrmsf*> value of the CP: logarithm plot of the average normalized root mean square fluctuation (*nrmsf*) value of a group, *ln*<*nrmsf* > vs. average expression value, $ln\langle \varepsilon^{k}\left(10\min\right)\rangle$ shows that *ln*<*nrmsf*>_CP_~ −2.5 in sandpile-type criticality, where $\varepsilon^{k}$ (10 min) represents the expression value of the *k*^th^ group (ordered from high to low *nrmsf* values) at *t* = 10 min and <…> represents the ensemble average of group expression. Grouping for (A) and (B) is ordered according to the fold-change in expression at 0–10 min. (**C**) Grouping $\left(c^{k}\left(t\right)\right)$ (CM grouping: Section 4.2.2) of whole-genome expression (baseline: CM($t_{j}$)) according to *nrmsf* reveals that this can be considered a fixed point and corresponds to the CM of whole-genome expression. This is true for both the population (**C**–**F**) and single-cell levels (**G**–**I**). *K* represents the number of groups with *n* number of elements ((**C**), (**D**): *n* = 891 mRNAs; (**E**), (**F**): *n* = 505 mRNAs; (**G**): *n* = 685 RNAs; (**H**): *n* = 666 RNAs; (**I**): *K* = 25, *n* = 525 RNAs; coloring: Section 4.1).

**Figure 3 ijms-21-04581-f003:**
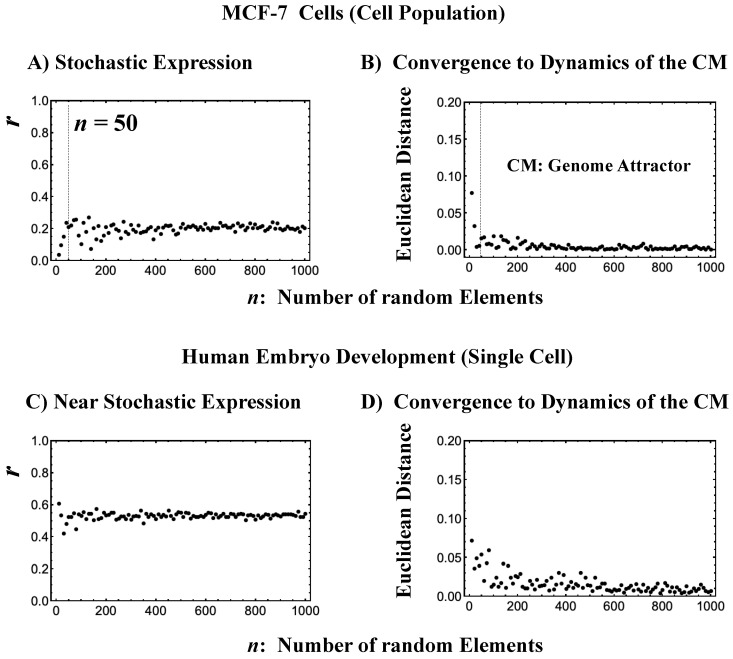
The center of mass (CM) acting as a genome attractor. MCF-7 cells (microarray data: (**A**) and (**B**)) and human embryo development (RNA-Seq data: (**C**) and (**D**)). The *x*-axis represents the number of randomly selected single-gene expression values in all four panels. Panels (**A**) and (**C**) illustrate that bootstrapping highlights a low pairwise Pearson correlation (*y*-axis) in randomly selected groups of expression (computed over experimental time points with 200 repetitions). This confirms the stochastic character of gene expression. Note: In (**C**), the removal of all genes with zero expression value at a specific cell state in RNA-Seq data does not change their correlation behavior (similar behavior is also observed in the mouse embryo case). Furthermore, the higher correlation in the embryo case makes the convergence to the CM, shown in (**D**), slower than in the cancer case (**B**). Panels (**B**) and (**D**) illustrate, in terms of Euclidean distance (*y*-axis; refer to the convergency of critical-state attractors: Section 2.6), the convergence of the dynamics of the CM of a randomly selected group of gene expression to that of the CM of whole-genome expression as the number of selected genes increases (average of 200 repetitions). The CP, the CM of whole-genome expression according to *nrmsf*, acts as the genome attractor (see Section 2.6 for heteroclinic attractors). Note: For the rest of the biological regulations throughout this report, similar behaviors are observed. The dashed lines in (**A**) and (**B**) indicate that coherent behaviors emerge at *n* = 50 single-gene expression values. Refer to experimental time points in Section 4.1.

**Figure 4 ijms-21-04581-f004:**
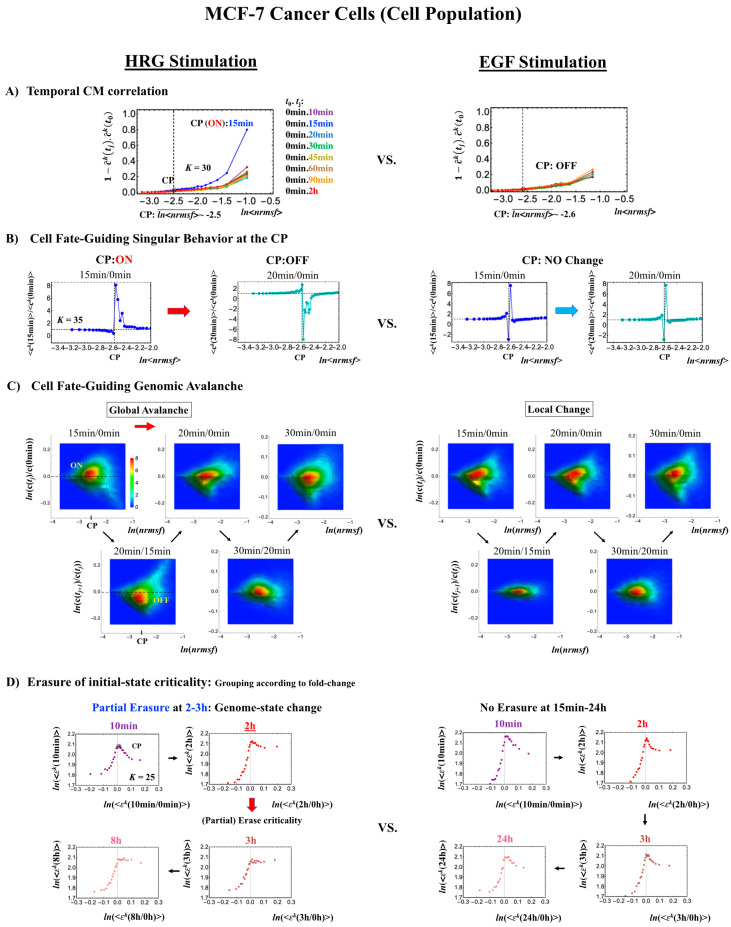
HRG- (left) and EGF-stimulated (right) MCF-7 cell-population responses. The CM correlation analyses reported in this figure are relative to gene groups ordered along their relative *nrmsf*: in (**A**), temporal CM correlation (*x*-axis; see Section 4.2.2): $1-{\hat{c}}^{k}\left(t_{j}\right).{\hat{c}}^{k}\left(t_{0}\right)$ (*y*-axis) is evaluated. At 15 min, the correlation response for HRG stimulation diverges from other responses, indicating that the CP is activated between 10 and 15 min for HRG, whereas for EGF stimulation, such a distinctly diverging correlation response does not occur (i.e., the CP is inactivated). Activation or inactivation of the CP marks the occurrence of cell differentiation for HRG and only proliferation (no differentiation) for EGF. Panels (**B**) and (**C**) provide direct evidence of the ON/OFF state of the CP. (**B**) The transition of the higher-order structure of the CP occurs for 15–20 min as shown in the fold-change from the initial time point. The swolled coil state (dominant positive fold-change: *y*-axis) at 15 min and compact globule (dominant negative fold-change) state at 20 min respectively correspond to the ON and OFF states of the CP (*K* = 35 groups, with each group containing *n* = 636 mRNAs). The sensitivity in singular behaviors at the CP is observed according to the number of groups. Throughout this report, to double the number of groupings, a new group is added to the plot, where the group is created by combining two halves from the nearest neighbors of the CM groups, and its average expression is evaluated. (**C**) The probability density function (PDF; see Section 4.2.3) of whole-genome expression (*z*-axis) shows that a global expression avalanche occurs at 15–20 min (coinciding with the change in the CP), where the maximum probability density occurs around the CP with a positive value of the natural log of fold-change (i.e., the CP is ON) at 15 min, whereas it becomes negative (OFF) at 15–20 min. (**D**) The (partial) erasure of initial-state criticality (grouping according to the fold-change in non-reference expression,$\;\varepsilon^{k}\left(t\right)$: *K* = 25: *n* = 891 mRNAs) occurs at 2–3 h for HRG (indication of genome-state change) and no erasure for EGF.

**Figure 5 ijms-21-04581-f005:**
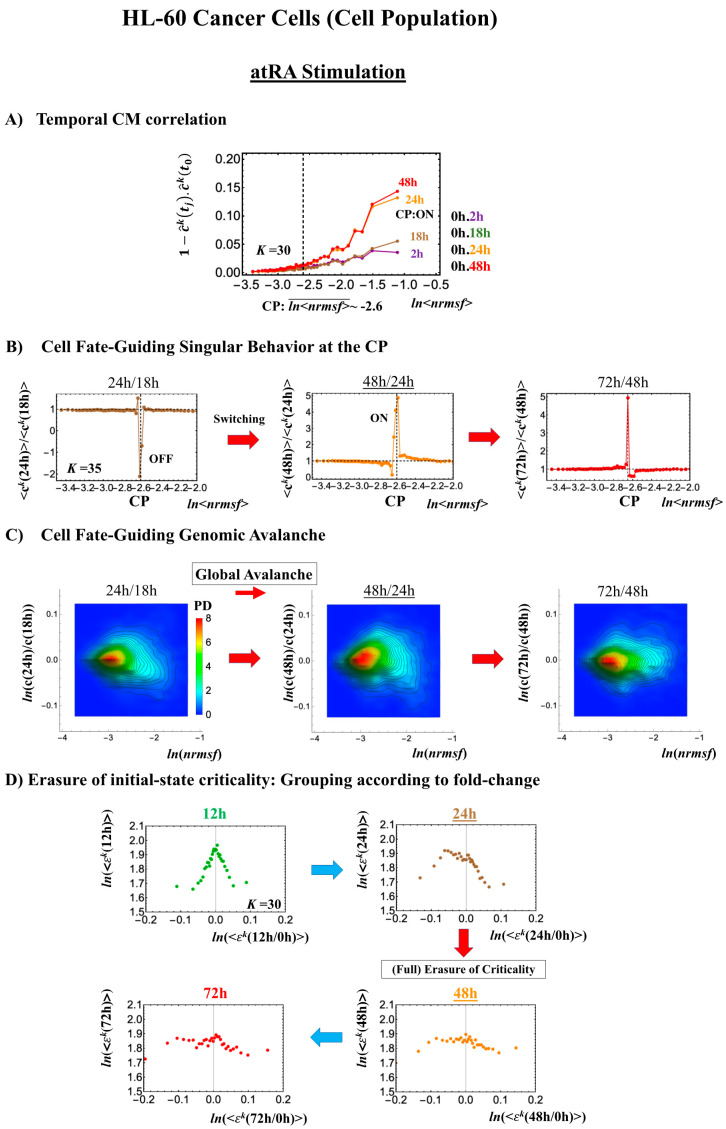
atRA-stimulated HL-60 cell population. (**A**) The divergent behavior of the temporal CM correlation with the initial whole-genome expression (*t* = 0) occurs at 24 h (*K* = 30: *n* = 420 mRNAs). This divergent response occurs for the ON state at 24 h indicated by (**B**) and (**C**). (**B**) Transition of the higher-order structure at the CP (*K* = 35: *n* = 360 mRNAs) is suggested to occur at 24–48 h, when initial-state criticality is erased (**D**). The switching of singular behaviors at the CP occurs with bimodal behaviors of the fold-change around the CP. The dominant negative fold-change indicates that the OFF state (compact globule DNA) occurs at 18–24 h, while the dominant positive fold change (swollen coil DNA) occurs at 24–48 h. (**C**) The temporal change in the PDF shows that a global avalanche occurs from 18–24 h (PDF: contracted) to 24–48 h (swollen) to 48–74 h (contracted), which supports the switching behaviors of the CP (**B**). (**D**) The full erasure of initial-state criticality occurs at 24–48 h, which indicates a cell-fate change. Underlined numbers represent the timing of the cell-fate change.

**Figure 6 ijms-21-04581-f006:**
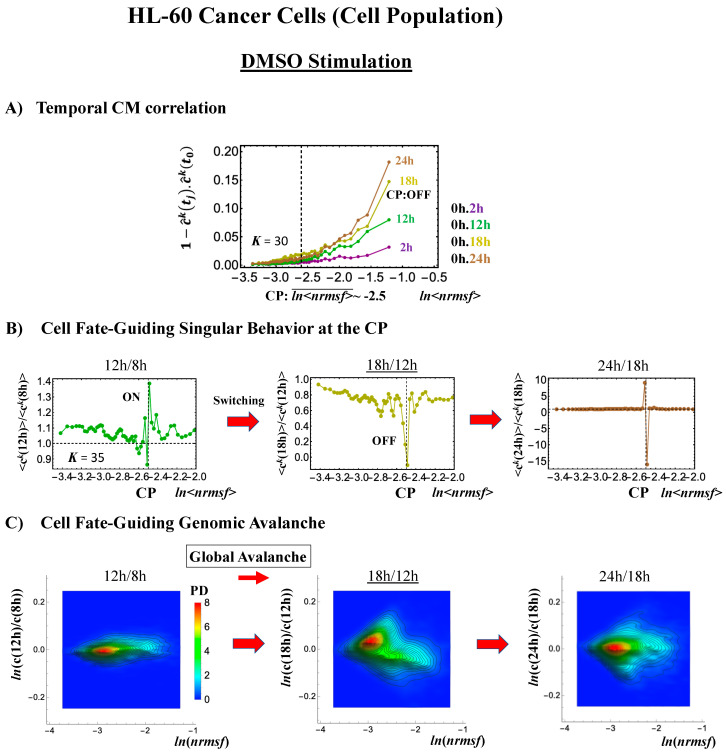
DMSO-stimulated HL-60 cell population. (**A**) The divergent behavior of the temporal CM correlation with the initial whole-genome expression (*t* = 0) occurs at 18 h (*K* = 30: *n* = 420 mRNAs). This positive divergent response occurs for the OFF state at 18 h indicated by (**B**) and (**C**) (refer to the cosine function, even-function around zero of two expression vectors; see Section 4.2.2). (**B**) Transition of the higher-order structure at the CP (*K* = 35: *n* = 360 mRNAs) from a swollen coil (ON) to a compact globule state (OFF) to a neutral state (neither ON or OFF; see (**C**)) at 18–24 h is suggested to occur at 12–18 h and 18–24 h for DMSO stimulation, when the initial-state criticality is erased (**D**). The sensitivity in singular behaviors at the CP is observed according to the number of groups. (**C**) The temporal change in the PDF shows that the probability density around the CP (*ln*<*nrmsf*> ~ −2.5) changes from positive to negative to zero, supporting the results of (**A**) and (**B**). This shows that a global avalanche occurs at 12–18 h and 18–24 h. (**D**) Full erasure happens at 8–12 h and 18–24 h for DMSO. The erasure of initial-state criticality occurs at 12–18 h, and thereafter, recovery of the initial-state criticality at 18 h indicates the existence of two SOC landscapes at 8–12 h and 12–18 h, respectively.

**Figure 7 ijms-21-04581-f007:**
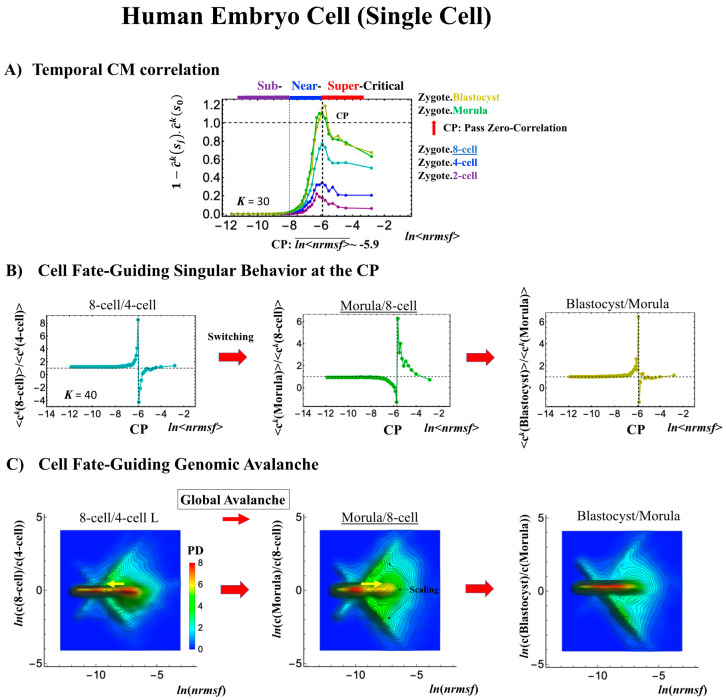
Human embryonic development (single-cell). The CP corresponds to (**A**) the peak of temporal CM correlation (see the difference from the cell population in Figure 4A and Figure 5A). A complete erasure of the zygote-CP (genome attractor) occurs after the 8-cell state (*K* = 30: *n* = 555 RNAs). Distinct response domains (critical states) as manifested in temporal CM correlations are shown (red: super-critical; blue: near-critical; purple: sub-critical), where the location of the CP lies at the boundary between near- and super-critical states. (**B**) Switching singular transition at the CP (*K* = 40: *n* = 416 RNAs) occurs at each cell state change (only shown from the 4-cell state here). The bimodal folding-unfolding feature at the CP suggests the occurrence of intra-chain segregation in the transition for genome-sized DNA molecules (see more in Section 3.2). (**C**) A genome avalanche occurs as a traveling density wave along low-expressed genes (around non-fold change: *y* = 0; the yellow arrow indicates the next direction of travel). Furthermore, three distinct scaling behaviors (linear behaviors in the density profiles in the log–log plot) emerge, which reveals that coordinated chromatin dynamics emerge to guide the reprogramming process. (**D**) The erasure of zygote criticality after the 8-cell state coincides with the timing of (**A**) the zero temporal CM correlation through (**B**) a switching singular transition at the CP. This timing further matches the timing of coherent perturbation of the genome engine passing through the non-equilibrium fixed point (see Section 2.6). Note: A linear behavior emerges in the grouping of randomly shuffled whole-genome expression, as expected; therefore, higher *nrmsf* is associated with higher expression (see more in Figure 3D in [7]).

**Figure 8 ijms-21-04581-f008:**
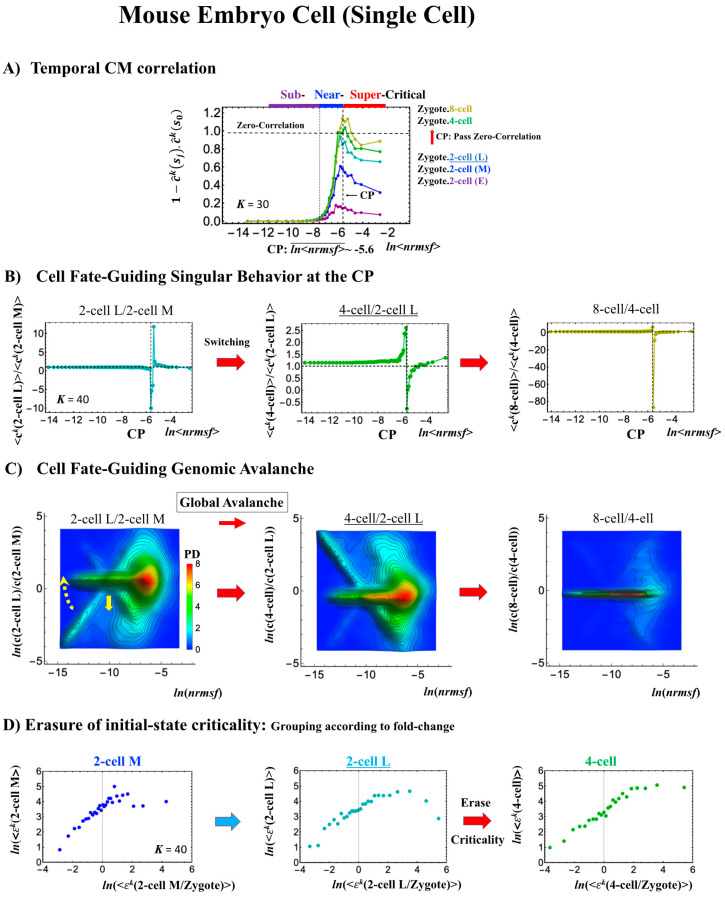
Mouse embryonic development (single-cell). Complete erasure of the zygote-CP (*K* = 30: *n* = 571 RNAs) occurs after the late 2-cell state (**B**) through the switching singular transition at the CP (*K* = 40: *n* = 428 RNAs). (**C**) A cell fate-guiding genome avalanche occurs from the middle 2-cell (2-cell M) to 4-cell states through the late 2-cell state (2-cell L), where the whole-genome PDF shifts from up- to downregulation and the scaling behavior (*ln*(*nrmsf*)< -11) reflects along the zero-fold change (*y* =0) to exhibit a coherent switching of folding and unfolding chromatin dynamics (see yellow solid and dashed arrows, respectively). The PDF shows that the CP is ON at the middle 2-cell – late 2-cell state, and OFF at the late 2-cell – 4-cell state and at the 4-cell state – 8-cell state. (**D**) Erasure of initial sandpile criticality occurs after the late 2-cell state, coinciding with the result of (**A**) and confirming the timing of the cell-fate change (after the late 2-cell state). Note: A traveling wave is also observed as seen in the human embryo cell (Figure 7C) between the zygote and other states.

**Figure 9 ijms-21-04581-f009:**
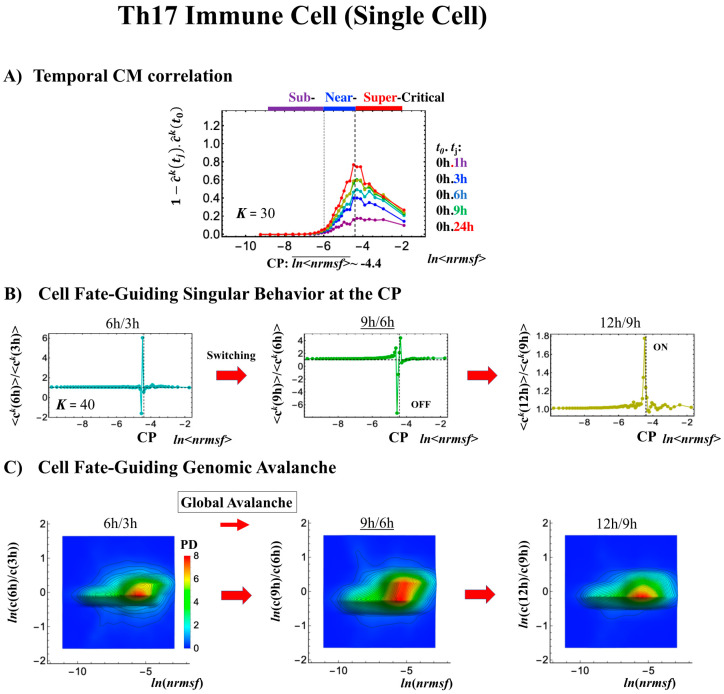
Differentiation of Th17 immune cells from Th0 cells (single-cell). (**A**) The CP features are the same as in the embryo cases; the only difference is that the CP never passes the zero-temporal correlation (*K* = 30, *n* = 437 RNAs). (**B**) The switching singular transition (*K* = 40: *n* = 328) at the CP (genome attractor) occurs from 3 to 9 h through 6 h, which induces (**C**) a large-scale (but not whole-genome as in the embryo cases) expression avalanche: swelling of the density profile from 3–6 h, and then contraction at 9–12 h. The timing of the cell-fate change (differentiation of Th17 cells from Th0) after 6 h is demonstrated by (**D**) erasure of the initial-state (*t* = 0) criticality. This timing is further supported by the same timing as in the switch from suppression to enhancement of the genome engine (see Section 3.1). The sensitivity of singular transition at the CP is observed according to the number of groups.

**Figure 10 ijms-21-04581-f010:**
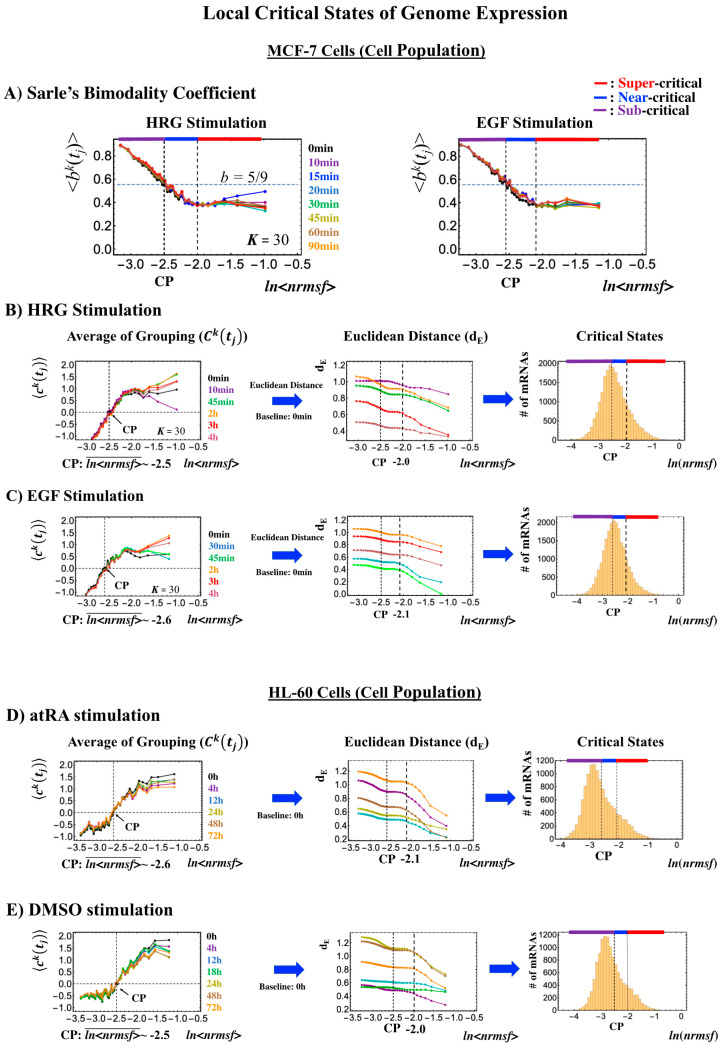
Systemic determination of local critical states for a cell population (microarray data). (**A**) In MCF-7 breast cancer cells, Sarle’s bimodality coefficient (*b* >5/9: onset of bimodal profile) shows that around the CP, a unimodal-to-bimodal transition (from high to low *nrmsf*) occurs for the group expression profile (*K* = 30: *n* = 742 mRNAs), which reveals three distinct profiles (red: super-; blue: near-; purple: sub-critical state) in whole-genome expression (left: HRG stimulation; right: EGF stimulation). To confirm these distinct behaviors (see panels (**B**) and (**C**)), which are based on the temporal response of the CM group (first column; refer to Figure 2), we examined the Euclidean distance (second column) between two ensemble averages, <$c^{k}\left(t=0\right)>$ and $<c^{k}\left(t_{j}\right)>$, (*k* = 1, 2,…,*K* = 25) from higher *nrmsf*, and confirmed three distinct behaviors with a boundary indicated by dashed vertical lines. Similarly, for HL-60 human leukemia cells, the corresponding results are shown for (**D**) atRA stimulation and (**E**) DMSO stimulation. In the third column (**B**–**E**), the corresponding region of critical states is shown in a histogram of gene expression according to *ln*(*nrmsf*). Note: The CP lies at the boundary between near- and sub-critical states, different from the case with a single cell (Figure 7, Figure 8 and Figure 9).

**Figure 11 ijms-21-04581-f011:**
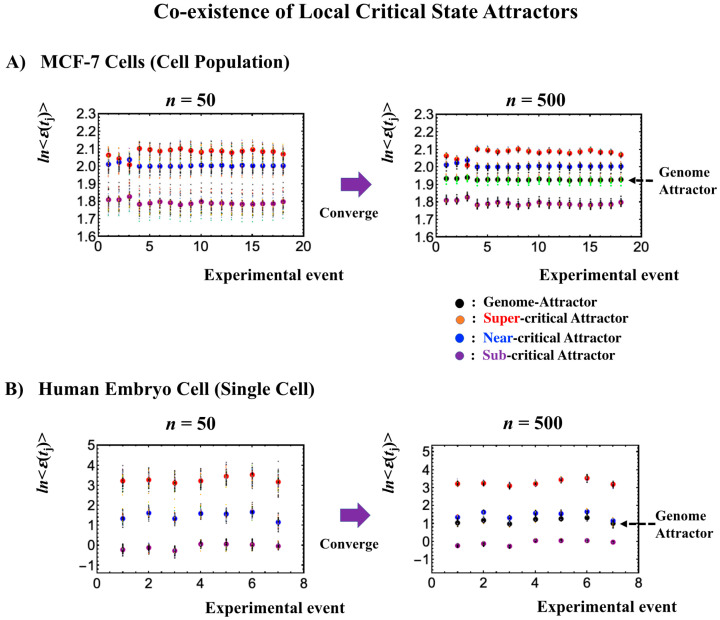
Local critical state attractors. Panels (**A**) for MCF-7 cells and (**B**) for human embryo cells illustrate that bootstrapping highlights that the convergence of the center of mass (CM) of randomly selected expression, *n* = 50 and *n* = 500 expressions, selected from a critical state converges to the CM of the critical state. For each random sampling, 40 repetitions were computed over experimental time points as depicted by each of the small circles. Large colored circles represent the CM of critical states (red: super-critical; blue: near-critical; purple: sub-critical state; see the number of expressions in local critical states in Table 1 and Table 2). The *x*-axis and *y*-axis correspond to experimental time points (Section 4.1) and the natural logarithm of the average of group expression, respectively. These confirm the existence of local critical states, and furthermore, their CM corresponds to local attractors (also confirmed in other biological regulations). Note: A mixture of critical states does not converge to the CM (see Figure 3 in [8]), showing that the genome attractor develops to form a heteroclinic critical-state attractor system.

**Figure 12 ijms-21-04581-f012:**
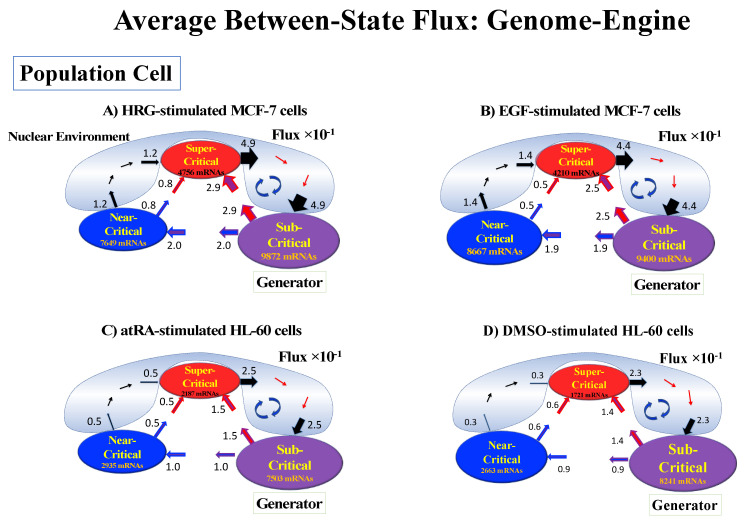
Genome engine mechanism revealed through average between-state expression flux. Cell population: (**A**) HRG-stimulated and (**B**) EGF-stimulated MCF-7 cancer cells; (**C**) atRA-stimulated and (**D**) DMSO-stimulated HL-60 cancer cells. Single-cell: (**E**) mouse and (**F**) human embryo development; (**G**) Th17 immune cell development. Figures show a common genome engine mechanism: sub-super cyclic flux forms a dominant flux flow to establish the genome engine mechanism. The sub-critical state acting as a “large piston” for short moves (low-variance expression) and the super-critical state acting as a “small piston” for large moves (high-variance expression) with an “ignition switch” (a critical point: the genome attractor) are connected through a dominant cyclic state flux as a “camshaft”, resulting in the anti-phase dynamics of two pistons (see the next Figure). Numerical values represent average between-state expression flux, whereas in the cell population, the values are based on a 10^−1^ scale.

**Figure 13 ijms-21-04581-f013:**
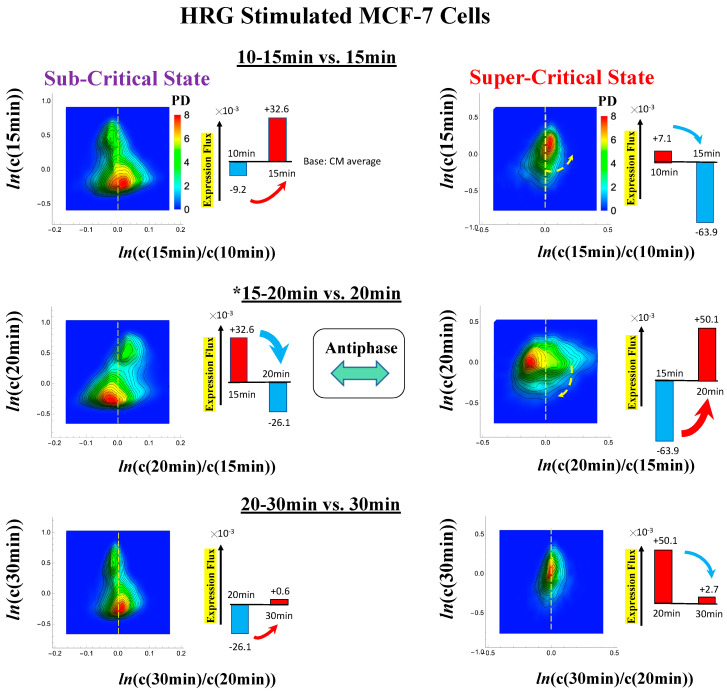
Dynamics of coherent behaviors around a phase transition in HRG-stimulated MCF-7 cells. The PDF (Section 4.2.3) reveals that the dynamics of coherent behaviors emerge in the stochastic gene expression in local critical states (sub-critical: left panel and super-critical states: right). This describes the anti-phase dynamics between the critical states and how sub-critical and super-critical attractors (Figure 11) dynamically self-organize stochastic gene expression as coherent oscillatory dynamics (see details in [6,7]). The vertical (*x*-axis) coherent oscillatory (up- and down-) regulations are explained by positive (obtaining flux) and negative (sending flux to other states) expression flux dynamics of critical states based on the genome engine (refer to Section 3.1). The biggest change in expression flux occurs between 15–20 min, which coincides with the timing of a global avalanche (Figure 4). The *x*- and *y*-axes represent the natural log of the fold-change and whole-genome expression vectors from the CM of critical states (Section 4.2.3), where whole-genome expression is ordered by the degree of *nrmsf* (see Figure 1).

**Figure 14 ijms-21-04581-f014:**
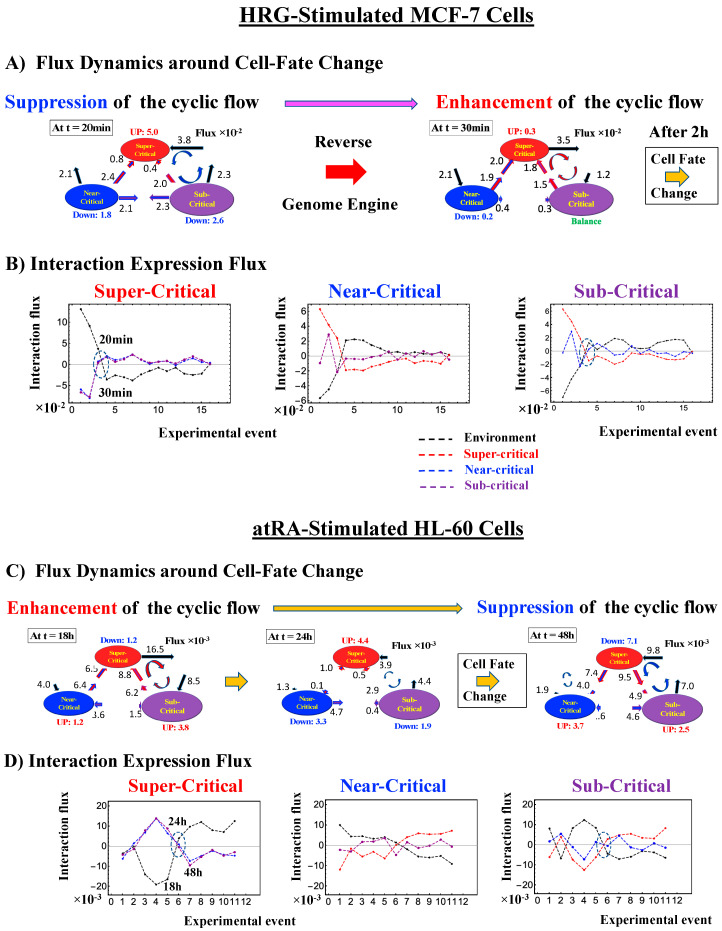
Cell population: mechanism of the cell-fate change through perturbation of the genome engine. (**A**) and (**B**) HRG-stimulated MCF-7 cells; (**C**) and (**D**) atRA-stimulated HL-60 cells, (**E**) and (**F**) DMSO-stimulated HL-60 cells. Before the cell-fate change, the interaction flux dynamics (indicated by the dashed oval: (**B**), (**D**), (**F**)) intersect each other. An interaction flux (red: super-critical state; blue: near- critical state; purple: sub-critical state; black: the nuclear environment) is measured from its average value (see details in Section 4.2.4). This intersection marks a non-equilibrium fixed point and the onset of the switch to coherent dynamics on the dominant cyclic flow (genome engine: Figure 12). The interaction flux dynamics show that, on the dominant cycle state flux, (**A**) the switch from suppression to enhancement for MCF-7 cells occurs before the cell-fate change (Figure 4**D**), and (**C**) and (**E**) the switch from enhancement to suppression for HL-60 cells occurs during the cell-fate change. Numerical values based on a 10^−2^ scale for MCF-7 cells and a 10^−3^ scale for HL-60 cells represent interaction flux. Refer to Section 4.1 for further details on experimental time points.

**Figure 15 ijms-21-04581-f015:**
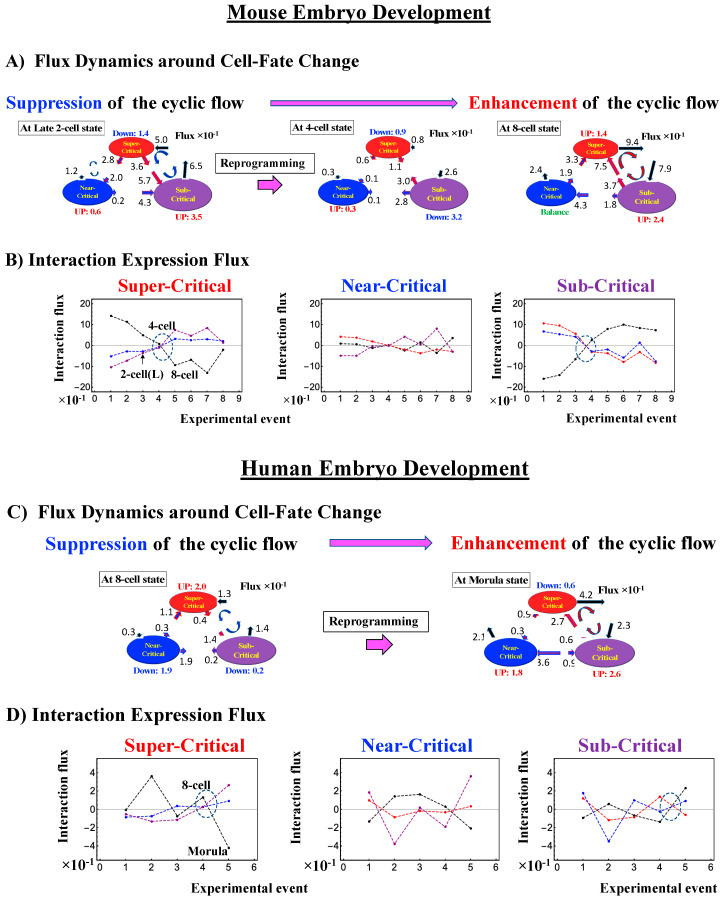
Single cell: mechanism of the cell-fate change through perturbation of the genome engine. (**A**) and (**B**) mouse embryo development; (**C**) and (**D**) human embryo development; (**E**) and (**F**) Th17 immune cell differentiation. As in cell-population responses, in the case of single cell, before reprogramming as well as cell differentiation, the interaction flux dynamics (the dashed oval: (**B**), (**D**), (**F**)) also intersect each other. Notably, reprogramming of embryo development (more evident in the mouse embryo) and cell differentiation in Th17 immune cells occurs through suppression–enhancement of the dominant cycle state flux between super- and sub-critical states. Numerical values based on 10^−1^ for embryo cells and 10^−2^ for Th17 cells represent between-state expression flux.

**Figure 16 ijms-21-04581-f016:**
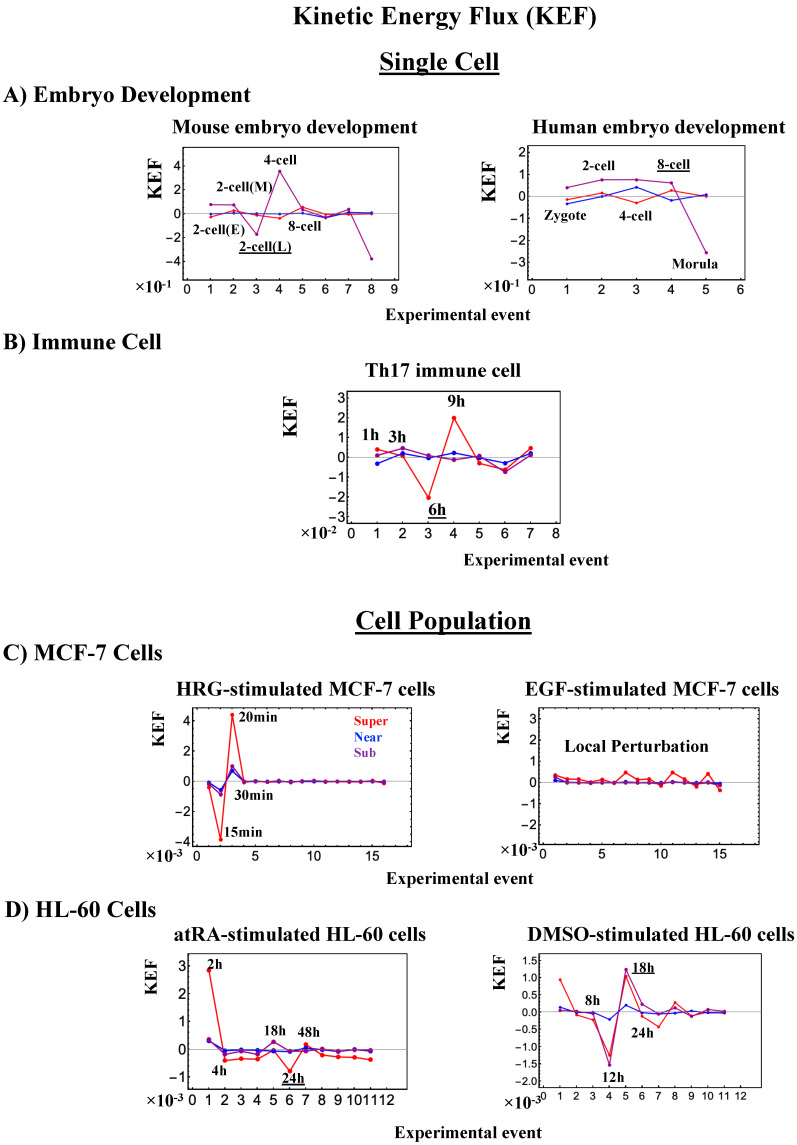
Kinetic energy flux (KEF). Global perturbation of the genome engine is well-manifested in the kinetic energy flux (Section 4.2.4) for single-cell responses in (**A**) embryo development and (**B**) immune cells, and population responses in (**C**) MCF-7 cells and (d) HL-60 cells. At the single-cell level, the timing of global perturbation, with the involvement of more than one critical state, coincides with that of the cell-fate change (Figure 7, Figure 8 and Figure 9). At the population level, kinetic energy flux is damping. In HRG-stimulated MCF-7 cells (**C**), the timing of the activation of the CP (ON at 15 min; Figure 4A: left panel) is within the global perturbation (10–30 min) (intersection of interaction flux at 20–30 min: Figure 14A), whereas in EGF stimulation (right panel; no cell differentiation [29]), only local perturbation occurs. On the other hand, in HL-60 cells (**D**), the timing of the perturbation of the genome engine (at 18 h, 24 h and 48 h) corresponds to the end of the damping (second-largest perturbation) for atRA-stimulated HL-60 cells (left panel) and the largest global perturbation (at 12 h, 18 h, and 24 h) for DMSO-stimulated HL-60 cells (right panel). Underlined numbers represent the timing just before the cell-fate change.

**Table 1 ijms-21-04581-t001:** Local critical states of single-cell genome expression. The boundaries of critical states with the number of RNAs are given for mouse (left column) and human (middle column) embryo development, and Th17 cell differentiation (right column), where the bold black value corresponds to the *ln*<*nrmsf*> value of the CP. * CP exits at the boundary between the super- and near-critical states.

Mouse Embryo Development	Human Embryo Development	Th17 Cell Differentiation
**Critical States:**	**Critical States:**	**Critical States:**
**Super-critical:** 3583 RNAs **−5.6 *** <*ln*<*nrmsf*>	**Super-critical**: 2607 RNAs **−5.9 *** <*ln*<*nrmsf*>	**Super-critical**: 3064 RNAs **−4.4 *** <*ln*<*nrmsf*>
**Near-critical**: 5922 RNAs −7.5< *ln*<*nrmsf*> < **−5.6**	**Near-critical**: 6707 RNAs −8.0 < *ln*<*nrmsf*> < **−5.9**	**Near-critical**: 4702 RNAs −6.0< *ln*<*nrmsf*> < **−4.4**
**Sub-critical**: 7636 RNAs *ln*<*nrmsf*> < −7.5	**Sub-critical**: 7345 RNAs *ln*<*nrmsf*> < −8.0	**Sub-critical**: 5368 RNAs *ln*<*nrmsf*> < −6.0

**Table 2 ijms-21-04581-t002:** Local critical states of cell-population genome expression. The boundaries of critical states with the number of mRNAs are given for HRG- and EGF-stimulated MCF-7 cells, atRA- and DMSO-stimulated HL-60 cells (from left to right column, respectively), where the bold black value corresponds to the *ln*<*nrmsf*> value of the CP. * CP exits at the boundary between the near- and the sub-critical states.

HRG-MCF-7 Cells	EGF-MCF-7 Cells	atRA-HL-60 Cells	DMSO-HL-60 Cells
**Critical States:**	**Critical States:**	**Critical States:**	**Critical States:**
**Super-critical**: 4756 RNAs −2.0 <*ln*<*nrmsf*>	**Super-critical**: 4210 RNAs −2.1 <*ln*<*nrmsf*>	**Super-critical**: 2187 RNAs −2.1 <*ln*<*nrmsf*>	**Super-critical**: 1721 RNAs −2.0 <*ln*<*nrmsf*>
**Near-critical**: 7649 RNAs **−2.5 *** < *ln*<*nrmsf*> <−2.0	**Near-critical**: 8667 RNAs **−2.6 *** < *ln*<*nrmsf*> < −2.1	**Near-critical**: 2935 RNAs **−2.6 *** < *ln*<*nrmsf*> < −2.1	**Near-critical**: 2663 RNAs **−2.5 *** < *ln*<*nrmsf*> <−2.0
**Sub-critical**: 9872 RNAs *ln*<*nrmsf*> < **−2.5**	**Sub-critical**: 9400 RNAs *ln*<*nrmsf*> < **−2.6**	Sub-critical: 7503 RNAs *ln*<*nrmsf*> < **−2.6**	**Sub-critical**: 8241 RNAs *ln*<*nrmsf*> < **−2.5**

**Table 3 ijms-21-04581-t003:** Single cell: state change in the CP associated with coherent perturbation of the genome engine. Table provides summary of when and how the state change in the CP with perturbation of the genome engine occurs for mouse and human embryo development, and Th17 cell differentiation (from top to bottom row, respectively). * Cell fate-guiding state change occurs in the CP. ** Reprogramming in mouse and human embryo development.

Biological Regulation	Location of the CP	State Change * in the CP	Coherent Perturbation of Genome Engine	Reprogramming **/Cell-Fate Cange
**Mouse Embryo Development**	*ln*<*nrmsf*>CP ~ −5.6	ON → OFF: At late 2-cell - 4-cell	Suppression- Enhancement	After late 2-cell **
**Human Embryo Development**	*ln*<*nrmsf*>CP ~ −5.9	Singular Switching: At 8-cell -Morula	Suppression- Enhancement	After 8-cell **
**Th17 Cell Immune** **Differentiation**	*ln*<*nrmsf*>CP ~ −4.4	OFF → ON: At 6–9 h	Suppression- Enhancement	After 6 h

**Table 4 ijms-21-04581-t004:** Cell population: state change in the CP associated with coherent perturbation of the genome engine. Table provides summary of when and how the state change in the CP with perturbation of the genome engine occurs EGF- and HRG-stimulated MCF-7 cells, DMSO- and atRA-stimulated HL-60 cells (from top to bottom row, respectively). * Cell fate-guiding state change occurs in the CP. ** Refer to a time lag mechanism in [10].

Biological Regulation	Location of the CP	State Change * in the CP	Coherent Perturbation of Genome Engine	Cell-Fate Change
**Cell Proliferation:** EFG-Stimulated MCF-7 Cells	*ln*<*nrmsf*>CP ~ −2.6	OFF State	Local Perturbation	NO
**Cell Differentiation:** HRG-Stimulated MCF-7 Cells	*ln*<*nrmsf*>CP ~ −2.5	OFF → ON: At 15–20 min	Suppression- Enhancement	After 2 h **
**Cell Differentiation:** atRA-Stimulated HL-60 Cells	*ln*<*nrmsf*>CP ~ −2.6	OFF → ON: At 24–48 h	Enhancement- Suppression	After 24 h
**Cell Differentiation:** DMSO-Stimulated HL-60 Cells	*ln*<*nrmsf*>CP ~ −2.5	ON → OFF: At 12–18 h	Enhancement- Suppression	After 18 h

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
