# Peer review of "Cell-Fate Determination from Embryo to Cancer Development: Genomic Mechanism Elucidated"

_ijms, 2020, doi:10.3390/ijms21134581_

Round 1
Reviewer 1 Report
The authors have addressed all the comments adequately.
Reviewer 2 Report
In this revised manuscript, the authors re-defined the concept of “cell fate change” in case specific context. They have also provided further explanation to illustrate the “critical point” in their data analysis. In summary, the revised manuscript provided sufficient information that answered all my questions.
This manuscript is a resubmission of an earlier submission. The following is a list of the peer review reports and author responses from that submission.
Round 1
Reviewer 1 Report
Major comments :
The introduction is too long and summarizes a past publication in too much depth. this can be easily summarized in a paragraph rather than writing a long review of past results.
it is not clear why the authors use specific cell types. for instance why did the authors use Th17 cells in their study as opposed to other immune cell types such Th1 and Th2 cells ?
How would these results differ if Th1 and Th2 cells were used in the study?
The discussion is very limited and almost seems like a recap of the results. The discussion needs to be revised substantially
The result section is formatted in a very confusing manner with several headings and sub-headings.
Minor comment
The first sentence in the abstract seems disconnected. Although i assume it is trying to state the objective of the study, it needs to be rewored to make that clear for the readers.
Reviewer 2 Report
In this manuscript, Tsuchiya. M and his colleagues described the development of mathematical models to infer cell fate change. Using available public source data, the authors examined common underlying machinery that potentially regulates cell fate change in cancer cells, immune cells and embryonic cells during development. Overall, the joint analysis using cells from cancer to embryonic cells is comprehensive and provide new insight into how cells are changing. However, the manuscript stays at its descriptive stage and seems quite preliminary to be published. The major points are as follows.
Major points:
- “Cell fate change” defines a switch from one determined cell type to another, for example, change from fibroblast to induced pluripotent cells, or differentiation of stem cell into neurons. The authors presented a misleading concept of “cell fate change” in current manuscript. Both MCF7 and HL-60 are cancer cell lines that will not change their cell identity after certain drug treatment. And the same with Th17 immune cells. The process described by the authors could be stated as cellular response instead of cell fate change. The tittle and contents of manuscript should be changed accordingly.
- Recent single cell RNA sequencing technology allows for the observation of transcriptomic profiling of cell fate change from fibroblast into other lineages including iPSCs, cardiomyocytes, and neurons. Use these source data may help to construct the “genome engine” to define the cell fate changes at deeper depth. The authors should use at least one of these examples to prove their theory.
- Identification of “critical point” as illustrated by the authors seems to be very important. Is there way to prove that these critical points are identified correctly? The authors should use experimental data, for example, the enriched gene sets, to show: 1) what is really a critical point, and 2) to demonstrate their definition matches with real biological process.